

# Classification and mapping of European fuels using a hierarchical-multipurpose fuel classification system

Elena Aragoneses[1], Mariano García[1], Michele Salis[2], Luís M. Ribeiro[3], and Emilio Chuvieco[1]

[1] Universidad de Alcalá, Departamento de Geología, Geografía y Medio Ambiente, Environmental Remote Sensing Research Group, Colegios 2, 28801 Alcalá de Henares, Spain

[2] National Research Council (CNR), Institute of BioEconomy (IBE), Traversa La Crucca 3, 07100 Sassari, Italy

[3] Univ Coimbra, ADAI, Department of Mechanical Engineering, Rua Luís Reis Santos, Pólo II, 3030-788

Coimbra, Portugal

**Correspondence.** Elena Aragoneses (e.aragoneses@uah.es)

**Abstract.** Accurate and spatially explicit information on forest fuels becomes essential to designing an integrated

fire risk management strategy, as fuel characteristics are critical for fire danger estimation, fire propagation and emissions modelling, among other aspects. This paper presents the conceptual development of a new fuel classification system that can be adapted to different spatial scales and used for different purposes. The resulting fuel classification system encompasses a total of 85 fuel types, that can be grouped into six main fuel categories (forest, shrubland, grassland, cropland, wet and peat/semi-peat land and urban), plus a nonfuel category. For the

forest cover, fuel types include two vertical strata, overstory and understory, to account for both surface and crown fires. Based on this classification system, a European fuel map at 1 km resolution, was developed within the framework of the FirEUrisk project, which aims to create a European integrated strategy for fire danger assessment, reduction, and adaptation. Fuels were mapped using land cover and biogeographic datasets, as well as bioclimatic modelling, in a Geographic Information System environment. The first assessment of this map was

performed by comparing it to high-resolution data, including LUCAS (Land Use and Coverage Area frame Survey) data, Google Earth images, Google Street View images, and the GlobeLand30 map. This validation exercise provided an overall accuracy of 88 % for the main fuel types, and 81 % for all mapped fuel types. Finally, to facilitate the use of this fuel dataset in fire behaviour modelling, a first assignment of fuel parameters to each fuel type was performed by developing a crosswalk to the standard fuel models defined by Scott and Burgan (FBFM,

Fire Behavior Fuel Models), considering European climate diversity.

**Key words.** Fuel maps, fire, risk, wildland, fuel types, FirEUrisk.

## 1 Introduction

Fire is a key disturbance factor for the dynamics (Thonicke et al., 2001; Pausas and Keeley, 2009) and distribution (Bond et al., 2005) of the vegetation ecosystems globally. Wildland fires affect forests' function (Bowman et al., 2009), structure (Koutsias and Karteris, 2003) and adaptation (Pausas and Keeley, 2009), while significantly contributing to emissions of greenhouse gases (Van Der Werf et al., 2017; Zheng et al., 2021), soil erosion (Shakesby, 2011), water and air pollution (Smith et al., 2011; Duc et al., 2018), and land cover change





(van Wees et al., 2021). Wildland fires also threaten human lives and properties and can cause important socio-economic impacts (Bowman et al., 2017, 2020).

Estimations based on coarse resolution satellite observations indicate that around 4 Mkm² (million km²) are globally burnt every year (Giglio et al., 2018; Lizundia-Loiola et al., 2020), although this evaluation is conservative, as they are based on global burnt area products that have shown to include significant omission errors

(Boschetti et al., 2019; Franquesa et al., 2022). The European territory is highly affected by wildland fires, which cause environmental, societal and economical losses (San-Miguel-Ayanz et al., 2020, 2021). In 2021, about 500,000 hectares were burnt in the European Union, from which 20 % affected Natura2000 and other protected sites, specially in Southern Europe. August was the worst month, including very large fires. Around 28 % of the total burnt area affected forest, and 25 % belonged to agricultural land types (San-Miguel-Ayanz et al., 2022). In

addition, global climate change will likely increase wildland fire risk and impacts in most of the European territory (Jones et al., 2019; IPCC, 2022). This justifies the necessity of improving the actual efforts to prevent and contain wildland fires in Europe (San-Miguel-Ayanz et al., 2021).

As it is well known, the Fire Environment defines the three key elements influencing fire initiation, propagation and effects: weather, topography and fuel (Countryman, 1972). Fire behaviour is highly dependent on

fuel (vegetation) characteristics, which is the only variable that can be managed to reduce fire propagation. In addition, fuel properties play a critical role in fire ignition and spread (Alvarado et al., 2020), as well as in the smouldering-flaming ratio of fire behaviour (Zheng et al., 2021), which in turn affects fire emissions.

Vegetation types with similar fire behaviour are grouped into fuel types and models (Pyne, 1984). The former indicate the classification of vegetation into categories with similar characteristics from a fire behaviour

perspective. The latter refer to the specific parameters required to model their fire behaviour (height, load, particle size, etcetera). Fuel types can refer to surface or canopy fuels. Forest understory and low vegetation formations are surface fuels, while elevated fuels, normally forest crowns, represent canopy fuels. Fire usually starts in surface fuels but may transfer to canopy fuels, causing crown fires, which are more dangerous than surface fires as they release more energy and propagate in larger fronts, being harder to control (Scott and Reinhardt, 2001).

Therefore, fuel type mapping is an essential tool in fire risk prevention, planning, and real-time fire management across multiple spatial scales (Keane et al., 2001) because it allows to spatially describe a key factor over which fire managers have control on (Keane and Reeves, 2012). Fire scientists require accurate and updated fuel maps to support fire strategic planning within a comprehensive fire danger assessment system. However, fuel mapping is challenging due to the high temporal and spatial variability of fuels (Keane et al., 2001).

In short, the starting point of fuel type mapping is to define the fuel classification system to be used, which includes the fuel types and models (parameters). Many fuel classification systems have been developed (Arroyo et al., 2008). All phases in their development process have heavily involved expert knowledge, from suppression specialists to researchers (Keane et al., 2001), because of the high diversity of fuels, their temporal and spatial variability and the lack of comprehensive fuel data across regions (Keane and Reeves, 2012).

The most commonly used fuel classification systems are the Northern Forest Fire Laboratory (NFFL) system (Anderson, 1982), the Fire Behaviour Fuel Models (FBFM) (Scott and Burgan, 2005), the Fuel Characteristic Classification System (FCCS) (Ottmar et al., 2007), all created for the United States; the Canadian Fire Behaviour Prediction System (Forestry Canada Fire Danger Group, 1992), and the Mediterranean-European Prometheus system (European Commission, 1999; Arroyo et al., 2008). Many of them include default parameters



and only refer to surface fuels, limiting their capability to prevent and manage crown fires (the most severe). Although they have been developed for specific regions and conditions, they have been widely used to map fuel types in other regions (García et al., 2011; Palaiologou et al., 2013; Marino et al., 2016; Aragoneses and Chuvieco, 2021).

Fuel types have been usually mapped through fieldwork, aerial photointerpretation, ecological modelling,
existing datasets and/or remote sensing (Arroyo et al., 2008). Remote sensing methods previously applied to fuel type mapping include a wide range of techniques and input data, from medium (Palaiologou et al., 2013; Alonso-Benito et al., 2013; Marino et al., 2016; Aragoneses and Chuvieco, 2021) to high spatial resolutions (Arroyo et al., 2006; Mallinis et al., 2008). Both passive (Alonso-Benito et al., 2013; Aragoneses and Chuvieco, 2021) and active (Riaño et al., 2003; González-Olabarria et al., 2012) sensors have been used, as well as a combination of sensors
(Mutlu et al., 2008; García et al., 2011; Palaiologou et al., 2013; Marino et al., 2016).

Fuel maps exist for continental scales, such as South America (Pettinari et al., 2014) and Africa (Pettinari and Chuvieco, 2015); and global scales (Pettinari and Chuvieco, 2016). However, in Europe, fuel mapping has been mostly developed for local and regional scales (Roulet, 2000; García et al., 2011; Stefanidou et al., 2020). The only European-level fuel cartography is the 2000 EFFIS fuel map (European Forest Fire Information System
(EFFIS), 2017), based on land cover and vegetation maps and using the NFFL system. Other works have mapped FBFM fuel models (Scott and Burgan, 2005) for the European subcontinental scale, such as the Iberian Peninsula (Aragoneses and Chuvieco, 2021).

The lack of an adapted-to-Europe fuel classification strategy is limiting since fuel models are site-specific and should be applied to the region for which they were developed to obtain the most realistic fuel mapping and
modelling (Arroyo et al., 2008). In this context, the ArcFuel project (Bonazountas et al., 2014) aimed in 2011-2013 to conceive a methodology to enable consistent fuel mapping production over Europe to support fire and emissions simulation scenarios, and the design of effective fire prevention and mitigation strategies. For this, it constructed a hierarchical vegetation fuel classification system adapted to Europe (Toukiloglou et al., 2013). Nevertheless, fuel cartography was only created for southern European national (Portugal and Greece), and
regional (Spain and Italy) scales and no European fuel map was generated (Bonazountas et al., 2014).

Considering the actual limitations of European fuel mapping, we aimed to three objectives. The first one was generating a European advanced fuel classification system to facilitate the integration of continental wildfire risk assessment, including both surface and canopy fuels. The new classification system should be hierarchical to facilitate scale integration, include both surface and canopy fuel types and be suitable for different purposes, from
fire behaviour simulation to fire emissions or fire danger assessment. The second objective was to develop a European fuel map at 1 km spatial resolution following the proposed fuel classification system. We aimed to develop a methodology that, combining expert knowledge, Geographic Information Systems (GIS), available datasets, and bioclimatic modelling, might be easily replicable and updated with low time and economic costs. Finally, the third objective was to assign fuel parameters to the derived fuel types, by relating them to existing fuel
models. We chose the FBFM standard fuel models (Scott and Burgan, 2005), as this system is widely used and very flexible. These three objectives (development of a fuel classification system, generation of a European fuel map and fuel parameterization) serve to organise the structure of this paper around three sections (Fig. 1). This work is expected to lay the framework for an integrated and homogeneous fire management strategy across

European countries. The present study is part of the FirEUrisk project, which aims to create a European integrated

strategy for fire danger assessment, reduction, and adaptation.

**1) FirEUrisk fuel classification system**

Development of the FirEUrisk hierarchical fuel classification system

**2) European fuel map**

Input data

Generation of the European fuel map

Resampling to target spatial resolution

Validation

**3) Fuel parameterization**

Crosswalk to standard fuel models

Fire Behaviour Fuel Models (FBFM)

**Figure 1.** General overview of the structure of this work.

**2 The FirEUrisk hierarchical fuel classification system**

**2.1 Development of the fuel classification system**

We developed the FirEUrisk hierarchical fuel classification system with three main requirements: it should be adapted to the great variety of European environmental conditions, include both surface and canopy fuels, and be suitable to work at different spatial scales. The main driver of the classification system was fire

behaviour modelling, but its use for fire risk assessment and fire emission estimations was considered as well. To define each of the fuel types, the land cover and vegetation descriptions of the UN-LCCS (United Nations Land Cover Classification System) from the UNESCO (United Nations Scientific and Cultural Organization) (UNESCO, 1973) and the FAO (Food and Agriculture Organization, 2000); and the European Environmental Agency's Corine Land Cover nomenclature (CLC) (Kosztra et al., 2019) were used. In addition to the mentioned

sources, for the wet and peat/semi-peat land fuel types, the definitions provided by the International Peatland Society (International Peatland Society, 2021) were also taken into account.

The FirEUrisk hierarchical fuel classification system used several criteria to discriminate fuel types. First, the main fuel cover, which differentiated six main fuel types: forest, shrubland, grassland, cropland, wet and peat/semi-peat land, and urban fuel types, plus a nonfuel category. For the forest fuel types, two vertical strata

were identified: the first-level referred to the overstory (canopy) characteristics, and the second-level to the understory characteristics. The former included three additional splitting criteria: leaf type (broadleaf/needleleaf), phenology (evergreen/deciduous), and fractional cover (open/closed). The latter included two aspects: understory type (grassland/shrubland/timber litter), and understory depth, that is, the height of the understory layer. For the



rest of the main fuel types, only one vertical stratum (first-level) was identified. For shrubland and grassland fuel types, subcategories were created based on fuel depth (height of the vegetation layer). For cropland, the split was based on cropland type (herbaceous/woody). For wet and peat/semi-peat land fuel subcategories, tree, shrubland and grassland formations were distinguished. For urban fuel types, the standard CLC division between continuous and discontinuous fabric was followed. For the nonfuel category, we distinguished water, snow, and ice; and bare soil and sparse vegetation, for high spatial resolutions.

## 2.2 The FirEUrisk hierarchical fuel classification system

The proposed hierarchical fuel classification system, FirEUrisk (Table A1 in Appendix A), encompassed a total of 85 fuel types for surface and canopy fuels, which were aggregated into six main fuel type categories, referred to the main fuel cover, which recall traditional land cover types, plus a nonfuel category. They were defined as follows:

- Forest: the tree cover is $\geq 15\%$ with a mean tree height $\geq 2$ m. Understory type refers to the fuel type in which the surface fire will spread in the forest.
- Shrubland: includes shrubs, scrub, garrigue, and maquis. It may have small trees $\leq 2$ m or tree cover $< 15\%$.
- Grassland: herbaceous non-cultivated vegetation. It may have small trees $\leq 2$ m or tree cover $< 15\%$.
- Cropland: cultivated vegetation (irrigated or not).
- Wet and peat/semi-peat land: it includes 1) Wetland: a permanent mixture of vegetation and water (salt, brackish, or fresh), including marshes; 2) Moorland/heathland: low and closed vegetation cover dominated by bushes, shrubs, dwarf shrubs and herbaceous plants, in a climax stage of development, including wet heath on humid or semi-peaty soils (peat depth $< 30$ cm), herbaceous vegetation, shrubs, and trees of dwarf growth $< 3$ m; 3) Peatland and peat bog: terrestrial wetlands in which flooded conditions prevent vegetation material from fully decomposing, which results in accumulation of decomposed vegetation matter and moss (peat), including valley, raised, blanket and quacking (floating) bogs with $> 30$ cm of peat layer, and mosses and herbaceous or woody plants within natural or exploited peat bogs; and 4) Moss and lichen.
- Urban: areas with $\geq 15\%$ built-up structures and/or buildings.
- Nonfuel: permanent water bodies, open sea, snow, ice, bare soil, sparse vegetation ($< 10\%$).

Fuel types subcategories were included to better estimate fuel models for each resulting fuel type category and would also lead to different fire behaviour. As previously indicated, two vertical strata were identified. The first-level identified the main vegetation cover, except for the forest fuel types, where it refers to the crown characteristics. The second-level referred to the understory and only applied to forest fuels. For the nonfuel categories, water/snow/ice and bare soil/sparse vegetation were also discriminated for the second-level. Discriminating all these subcategories may be quite challenging and should be adapted to the working scale of the fuel type product and, accordingly, to the quality of the input data available to produce it. The fuel type categories of the first-level (Table 1) should be more suitable for continental or global fuel products, while the second-level should be better adapted to local or regional studies, where more detailed information can be available. In this paper, the European fuel type dataset was based on the first-level of the classification hierarchy. The area includes 33 countries and covers around 5 Mkm$^2$ of land. The spatial resolution for the target area is 1 km. This product



was developed to help the strategic planning of fire management in Europe, as part of the research activities of the FirEUrisk project, although we encourage its use in other projects and applications.

**Table 1.** 24 first-level FirEUrisk fuel types expected to be mapped at continental scale. See Table A1 in Appendix A for the complete FirEUrisk fuel classification system.

| FirEUrisk fuel type | | FirEUrisk fuel type | |
| --- | --- | --- | --- |
| Code | Description | Code | Description |
| 1111 | Open broadleaf evergreen forest | 23 | High shrubland [$\geq$ 1.5 m] |
| 1112 | Closed broadleaf evergreen forest | 31 | Low grassland [0-0.3 m] |
| 1121 | Open broadleaf deciduous forest | 32 | Medium grassland [0.3-0.7 m] |
| 1122 | Closed broadleaf deciduous forest | 33 | High grassland [ $\geq$ 0.7 m] |
| 1211 | Open needleleaf evergreen forest | 41 | Herbaceous cropland |
| 1212 | Closed needleleaf evergreen forest | 42 | Woody cropland |
| 1221 | Open needleleaf deciduous forest | 51 | Tree wet and peat/semi-peat land |
| 1222 | Closed needleleaf deciduous forest | 52 | Shrubland wet and peat/semi-peat land |
| 1301 | Open mixed forest | 53 | Grassland wet and peat/semi-peat land |
| 1302 | Closed mixed forest | 61 | Urban continuous fabric |
| 21 | Low shrubland [0-0.5 m] | 62 | Urban discontinuous fabric |
| 22 | Medium shrubland [0.5-1.5 m] | 7 | Nonfuel |

## 190 3 The European fuel map

### 3.1 Study area

The study area is the European territory as defined by the FirEUrisk project, with around 5 Mkm$^2$ of land (Fig. 2). The most historically affected European countries by wildland fires have been Portugal, Spain, Italy, Greece, and France. However, a recent increase in fire activity in northern Europe has been observed (e.g., fires in 195 Sweden in 2018). The most dangerous fire conditions in the European territory are usually observed during the summer months, the peak of the fire season in the most affected European Union countries, although a high number of fires also occur in winter, spring, and autumn (San-Miguel-Ayanz et al., 2021).





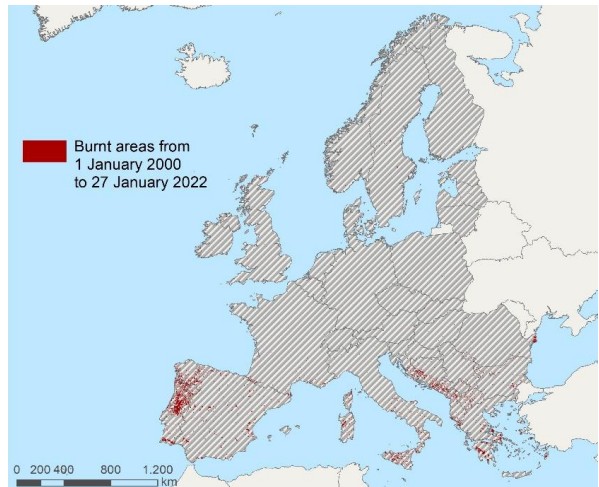

**Figure 2.** Study area and burnt areas from 1 January 2000 to 27 January 2022 (EFFIS, 2021).

### 3.2 Methods

### 3.2.1 Input data

The generation of the European fuel map with the targeted first-level fuel types (Table 1) was based on the combination of existing land cover and biogeographic regions datasets covering European territory and bioclimatic models.

Due to the similarity between the fuel types of the FirEUrisk fuel classification system and the 2019 discrete Copernicus Global Land Cover map (Copernicus GLC map) legend (Buchhorn et al., 2020), this land cover dataset was used as base cartography for the generation of the European fuel map. The Copernicus GLC map has 100 m resolution and is based on PROBA-Vegetation (PROBA-V) sensor (Buchhorn et al., 2020) with an overall accuracy of 79.9 % for continental land cover categories and 72.8 % for regional land cover categories over Europe (Tsendbazar et al., 2020). Whenever the land cover information of the Copernicus GLC map was insufficient to map a FirEUrisk fuel type, we used the three following input datasets to derive the required information:

1) the 2020 global Climate Change Initiative Land Cover map (CCI LC map) at 300 m resolution based on Medium Resolution Imaging Spectrometer (MERIS), PROBA-V and Sentinel-3 Ocean and Land Colour Instrument (OLCI) (Copernicus Climate Change Services, 2020) with an overall accuracy of 70.5 % (Defourny et al., 2021);

2) the 2018 pan-European Corine Land Cover raster map (CLC map) at 100 m resolution based on photointerpretation of Sentinel-2 MultiSpectral Instrument (MSI) and Landsat-8 Thematic Mapper (TM) images (European Union Copernicus Land Monitoring Service, 2018), with an overall accuracy of 92.67 % (European Union Copernicus Land Monitoring Service, 2021);

3) the 2019 fraction cover Copernicus Global Land Cover map at 100 m resolution for the built-up category (Built-up fraction cover Copernicus GLC map) (Buchhorn et al., 2020) based on the 2015 World Settlement Footprint map (Marconcini et al., 2020) and yearly-updated OpenStreetMap images with a mean absolute error of 0.8 % (Tsendbazar et al., 2020).



The Copernicus GLC map (Buchhorn et al., 2020) and the Built-up fraction cover Copernicus GLC map (Buchhorn et al., 2020) were downloaded in tiles for the study area and mosaicked. All input datasets were reprojected to ETRS89 Lambert Azimuthal Equal Area using the nearest neighbour method and with the same spatial resolution as the Copernicus GLC map. The input datasets were also clipped to the study area.

Also, to account for fuel depth categories (low, medium, and high shrubland and grassland fuel types), we used datasets and bioclimatic models (Saglam et al., 2008; Smit et al., 2008; Fick and Hijmans, 2017; Bohlman et al., 2018; Zhang et al., 2018a) to relate environmental conditions with fuel depth.

To account for bioclimatic variations across Europe we used the 2016 dataset of Europe's biogeographic regions by the EEA (European Environment Agency, 2016). The study area had nine biogeographic regions: Alpine, Arctic, Atlantic, Black Sea, Boreal, Continental, Mediterranean, Pannonian and Steppic. For each biogeographic region, we analysed climate graphs from 1861 to 2019 of several representative cities using the ClimateCharts.net platform (Zepner et al., 2020). The biogeographic regions whose climate graphs presented at least one dry summer month were assigned to the arid/semi-arid regime. A dry summer month is interpreted as a month whose sum of monthly precipitation (mm year$^{-1}$) is less than twice the mean month temperature (ºC) (Zepner et al., 2020). The biogeographic regions not meeting this condition were assigned to the sub-humid/humid regime. The final general bioclimatic regimes were rasterized to 100 m and 1 km resolution using the maximum area method.

### 3.2.2 Generation of the European fuel map

Methods to generate the European fuel map are summarised in Fig. 3.

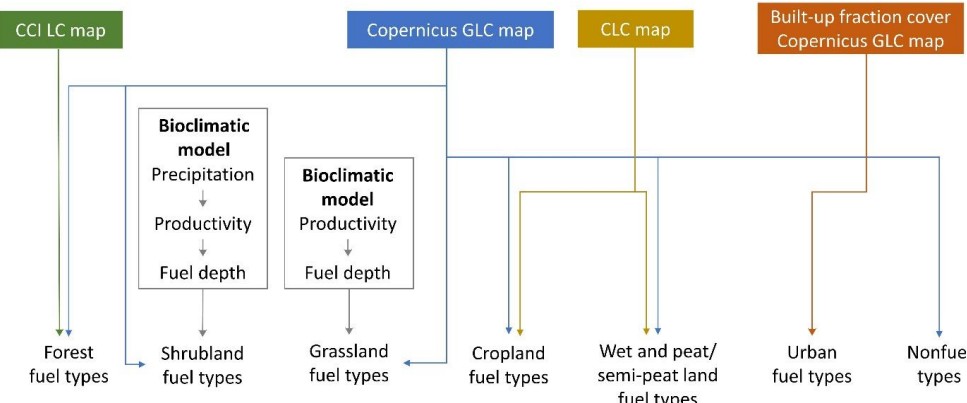

**Figure 3.** Methodology used to generate the European fuel map. Sources are in the text.

A) Forest fuel types

Information on the leaf type, phenology, and fractional cover of forest fuels was obtained from the Copernicus GLC map (Buchhorn et al., 2020). This dataset defines all the first-level forest fuel types in the FirEUrisk fuel classification system, plus two more categories only referring to fractional cover: unknown open forest and unknown closed forest. Pixels falling in these two categories were overlapped with the CCI LC map (Copernicus Climate Change Services, 2020), previously resampled from 300 to 100 m using the nearest neighbour



method to match the resolution of the Copernicus GLC map. This allowed determining the leaf type (broadleaf/needleleaf) and phenology (evergreen/deciduous) of the unknown forest. The pixels identified as unknown forest in the Copernicus GLC map but not as forest in the CCI LC map were assigned the category of the CCI LC map.

B) Shrubland fuel types

The shrubland cover was extracted from the Copernicus GLC map (Buchhorn et al., 2020). To our knowledge, no global or European datasets on shrubland fuel depth, which is the height of the shrubland layer, are available. This variable is quite important, as shrubland depth is directly related to shrubland productivity (Radloff and Mucina, 2007; Saglam et al., 2008; Ali et al., 2015), which is mainly determined by the Mean Annual Precipitation (MAP) (Shoshany and Karnibad, 2015; Paradis et al., 2016; Bohlman et al., 2018; Zhang et al., 2018b) through biomass accumulation (Keeley and Keeley, 1977; Schlesinger and Gill, 1980; Gray and Schlesinger, 1981; Bohlman et al., 2018). This is especially relevant in the arid/semi-arid regime, like the Mediterranean (Shoshany and Karnibad, 2011). Therefore, shrubland fuel depth was obtained from a bioclimatic model adapted to arid/semi-arid conditions with three steps: first, mapping European MAP; second, estimating shrubland productivity from MAP; and third, estimating shrubland fuel depth from productivity.

Global 1970-2000 MAP at 1 km resolution was downloaded from WorldClim 2 dataset (Fick and Hijmans, 2017). The data were reprojected from WGS84 Geographic latitude/longitude to ETRS89 Lambert Azimuthal Equal Area using the bilinear method and clipped using the European shrubland mask.

The estimation of shrubland productivity was based on a linear model (Eq. 1) that related shrubland productivity and MAP for California (Bohlman et al., 2018). This model was derived from a literature review, and Californian bioclimatic conditions are similar to those of European arid/semi-arid regions, as can be checked in the ClimateCharts.net platform (Zepner et al., 2020). Therefore, it was used to calculate the mean potential shrubland productivity for each pixel.

$$\text{Biomass (g m}^{-2}\text{)} = 9.6696 \text{ MAP (mm year}^{-1}\text{)} - 1301.7 \tag{1}$$

Finally, we used a linear empirical model (Eq. 2) that related shrubland fuel depth and productivity for two study areas in Turkey (Saglam et al., 2008) that are similar to European conditions: 650 and 1200 mm year$^{-1}$ mean precipitation. We applied this model to estimate shrubland fuel depth, constraining the outputs to the [0-6] m range. Last, each shrubland fuel depth pixel was assigned to its corresponding shrubland group of the FirEUrisk fuel classification system.

$$\text{Depth (m)} = ((\text{Biomass (g m}^{-2}\text{)} / 1000) - 0.708) / 2.8 \tag{2}$$

C) Grassland fuel types

The Copernicus GLC map (Buchhorn et al., 2020) was used to identify grassland areas. To our knowledge, no global or European datasets on grassland fuel depth, that is, the height of the grassland layer, are available. Grassland depth is directly related to grassland productivity (Zhang et al., 2018a; Crabbe et al., 2019; Michez et al., 2019; Batistoti et al., 2019) which correlates with environmental conditions (Smit et al., 2008),



mainly the MAP: regions with more precipitation have higher grasslands with higher productivity (Smit et al., 2008; Nunez, 2019; Neal, 2021). The most productive grasslands are located in central Europe, while lower grasslands are located in the Mediterranean and Arctic regions (Smit et al., 2008). Information on the grassland fuel depth was obtained from a bioclimatic model with two steps: first, mapping grassland productivity, and second, estimating grassland fuel depth from productivity.

First, European grassland productivity was derived from the consistent inventory of regional statistics (Smit et al., 2008) for the European environmental zones (Metzger et al., 2005), similar to the European biogeographic regions. The mean grassland productivity values were assigned to each polygon of the biogeographic regions' map and were subsequently rasterized using the maximum area method to 100 m resolution, representing the European mean grassland productivity by biogeographic region. The map was then clipped by the grassland mask to obtain this information for the grassland pixels.

Second, to estimate European grassland fuel depth, we used a linear empirical model (Eq. 3) that relates grassland depth and biomass for China (Zhang et al., 2018a). We considered this model appropriate for Europe because Chinese grasslands are also generally temperate and the model was developed considering three study areas that relate to European conditions: 1) 80-220, 2) 600, and 3) 850-1000 mm year$^{-1}$ mean precipitation. With this model, we estimated grassland fuel depth for every pixel. Finally, each pixel was assigned to a FirEUrisk grassland group according to fuel depth. Outliers (pixels with < 0 m) were reclassified to 0 m.

$$\text{Depth (m)} = (\text{Biomass (g m}^{-2}) - 161.09) / 578.3 \tag{3}$$

D) Cropland fuel types

The herbaceous cropland cover was extracted from the Copernicus GLC map (Buchhorn et al., 2020), as this dataset only has information on this type of cropland cover. The CLC map (European Union Copernicus Land Monitoring Service, 2018) was overlapped with the previous map to extract the location of the woody cropland pixels (CLC categories: 221, 222, 223).

E) Wet and peat/semi-peat land fuel types

The Copernicus GLC map (Buchhorn et al., 2020) was used to extract the location of the wetland-herbaceous cover, as this dataset only has information on this type of wetland cover; and the moss and lichens cover. These categories were assigned to the grassland wet and peat/semi-peat land fuel type. Then, the CLC map (European Union Copernicus Land Monitoring Service, 2018) was used to extract the pixels of the peatland and moorland/heathland categories (CLC categories: 322, 412). These pixels were overlapped with the Copernicus GLC map to classify them into tree, shrubland or grassland wet and peat/semi-peat land fuel types, according to the cover type from the Copernicus GLC map they overlapped.

F) Urban fuel types and nonfuel types

The Built-up fraction cover Copernicus GLC map (Buchhorn et al., 2020) was used to extract the location of the pixels with ≥ 15 % and ≥ 80 % of urban cover. Pixels with ≥ 80 % of urban cover were assigned to urban continuous fabric and the rest of the identified urban pixels were assigned to urban discontinuous fabric.

The permanent water bodies, open sea, snow and ice, and bare/sparse vegetation (< 10 %) categories from the Copernicus GLC map (Buchhorn et al., 2020) were reclassified to the nonfuel category.

### 3.2.3 Resampling to the target spatial resolution

After obtaining the first draft of the fuel type dataset at 100 m resolution, it was reprojected to the target spatial resolution of 1 km. Before resampling, potential noise in the cross-tabulation process was minimised by using a majority filter. We performed filtering tests using 3 x 3, 5 x 5, and 7 x 7 moving windows and chose the most suitable according to a balance between information preservation and noise removal.

Resampling was carried out using a custom method that accounts for the spatial heterogeneity of
European landscapes. First, the dominant categories within each 1 km$^2$ pixel were estimated by computing the frequency of the fuel type categories within the 10 x 10 pixels contained in each 1 km$^2$. The base resampling criterion was to choose the dominant (first-mode) category within the target pixel. However, to produce a more accurate resampled fuel map that considered complex landscape covers (e.g., mixed forest) and the most dangerous fuel type between two similar co-dominant fuel types; the combination of categories in Table 2 was performed
whenever there were two co-dominant categories with the same frequency in a group of 10 x 10 pixels, or the frequency of the co-dominant (second-mode) category was higher than half the frequency of the dominant category (first-mode). The combination of the categories in Table 2 was carried out regardless of which category was dominant and which co-dominant. In the case of a combination of co-dominant categories not included in Table 2, the resampling was performed by randomly choosing one of the co-dominant categories. After resampling, the
number of first-mode categories within the 10 x 10 pixel groups was calculated to check the adequacy of the smoothing and resampling method to the data.

**Table 2.** Combination of fuel types to resample the 100 m resolution European fuel map to the target 1 km spatial resolution.

| Original fuel map (100 m) | | Target fuel map (1 km) |
|---|---|---|
| **Category A** | **Category B** | **Resampling category** |
| Broadleaf forest | Needleleaf forest | Mixed forest |
| Evergreen forest | Deciduous forest | Mixed forest |
| Mixed forest | Any other type of forest | Mixed forest |
| Open forest | Closed forest | Open forest |
| Urban continuous fabric | Urban discontinuous fabric | Urban discontinuous fabric |
| Forest | Shrubland | Shrubland |
| Forest | Grassland | Grassland |
| Shrubland | Grassland | Grassland |
| Low shrubland | Medium shrubland | Medium shrubland |
| Low shrubland | High shrubland | Medium shrubland |
| Medium shrubland | High shrubland | High shrubland |
| Low grassland | Medium grassland | Medium grassland |
| Low grassland | High grassland | Medium grassland |
| Medium grassland | High grassland | High grassland |



### 3.2.4 Validation methods

We performed a two-step validation of the final European fuel map at 1 km resolution. Considering the infeasibility of ground validation of the final product, we first carried out validation for the six main fuel types (forest, shrubland, grassland, cropland, wet and peat/semi-peat land, and urban) of our classification, plus the nonfuel category, using LUCAS (Land Use and Coverage Area frame Survey) as reference data. LUCAS points are derived from a systematic survey, performed every three years by Eurostat to identify land cover and use changes (including photos) in the European Union (Eurostat, 2022a). 2018 LUCAS microdata for Europe were downloaded (Eurostat, 2022b), reprojected from WGS84 Geographic latitude/longitude to ETRS89 Lambert Azimuthal Equal Area, and mapped by their observation coordinates.

Selection of suitable LUCAS points for the main fuel types validation was based on the following criteria: no GPS accuracy issues, field survey with point visible < 100 m and observation on the point, parcel area ≥ 10 ha, 100 % land cover coverage, not referring to small features (roads, railway, pipelines, telecommunications, etcetera) because these elements occupy a small fraction of a 1 km$^2$ pixel and are not identified in a fuel type product at this resolution, and photo on point. We selected only those LUCAS points with available photos, so our fuel types associated with fuel depth or multilayer structure could be estimated visually. Moreover, to avoid border effects and make LUCAS points more comparable to our target spatial resolution (1 km), they should be located within large homogeneous areas. So, LUCAS points were buffered 200 m and only those points whose buffers met these three conditions were kept: 1) falling 88.5 % inside a polygon ≥ 4 km$^2$ of the 100 m vectorised fuel map, 2) falling completely inside a polygon of the 1 km$^2$ vectorised fuel map for the main fuel types, and 3) falling completely inside the study area. We used 88.5 % instead of 100 % to have enough pixels to perform validation for all main fuel types. Finally, we obtained 28,240 suitable LUCAS points, whose land cover categories were reclassified to the most similar FirEUrisk main fuel types and used to generate 5,016 validation points by stratified random sampling. A confusion matrix was computed for quantitative analysis.

After validating the main fuel types from this automatic procedure, we performed a second validation exercise, aiming to assess all mapped fuel types, which required to obtain reference information on leaf type, phenology, fractional cover, fuel depth, and type. Since this required a visual interpretation, a 20 % subset of the 5,016 validation points was selected by stratified random sampling. Each point was assigned to a fuel category by visual interpretation of four information sources: 1) the latest Google Earth images to observe the 1 km$^2$ pixel, 2) Google Street View images, 3) 2018 LUCAS photos at a maximum distance of 200 m, and 4) the 2020 global land cover GlobeLand30 map (30 m resolution) (Chen and Ban, 2014) with 85.72 % of overall accuracy, based on Landsat and Huanjing (HJ-1) images to help to validate forest and urban covers. The GlobeLand30 tiles for the European territory were downloaded (http://www.globallandcover.com), mosaicked, and reprojected from WGS84 Geographic latitude/longitude to ETRS89 Lambert Azimuthal Equal Area using the nearest neighbour method. We generated binary layers for forest and urban covers and computed the percentage of each cover within each 1 km$^2$ pixel of the final European fuel map. Some fuel types with low representation in Europe had an insufficient number of pixels with suitable LUCAS points. To analyse at least 10 pixels of each fuel type, we also used LUCAS points not matching all quality criteria for those fuel types. Quantitative analysis through a confusion matrix was performed.

Finally, the two confusion matrices (one for the main fuel types, another for all mapped fuel types) were compared to the results obtained from the validation of the 2015 Copernicus GLC map over Europe (Tsendbazar



et al., 2020) to check if accuracy values were similar. We used the 2015 map instead of the 2019 one, because the
confusion matrix of the 2019 map was not available. This was considered reasonable as categories' accuracies
show consistency between the 2015 and 2019 Copernicus GLC maps varying less than 2 % and being the stability
index < 15 % for most categories, except for herbaceous wetlands, whose producer accuracy increased and user
accuracy decreased between 2015 and 2019 (Tsendbazar et al., 2021).

### 3.3 Results

Based on the analysis of bioclimatic conditions, the European Black Sea, Mediterranean and Steppic
biogeographic regions were assigned to the arid/semi-arid regime (19.83 % of the territory, in southern Europe);
and the European Alpine, Arctic, Atlantic, Boreal, Continental, and Pannonian biogeographic regions were
assigned to the sub-humid/humid regime (80.17 % of the territory, in central and northern Europe) (Fig. B1 in
Appendix B).

The application of the bioclimatic models for estimating the shrubland and grassland fuel depth in the
European fuel map at 100 m resolution, yielded the distribution of these fuel types' depth in Europe. Medium and
high shrubland predominate in Europe with 2.28 % of the shrubland fuel types being low, 51.80 % medium, and
45.92 % high. The grassland fuel depth representation is similar for all groups: 35.81 % of the grassland fuel types
are low, 31.94 % are medium, and 32.25 % are high, being the maximum grassland fuel depth 1 m approximately
(Fig. 4).

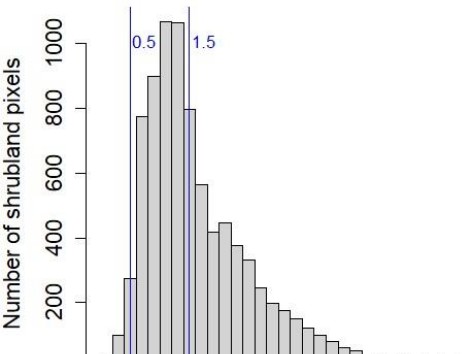

**Figure 4.** Histograms for shrubland and grassland fuel depth (m) in Europe. The blue lines represent the fuel
depth threshold used to subdivide shrubland and grassland fuel types.

The European fuel map at 100 m resolution is an intermediate result. It included 22 first-level fuel types
in Europe. Forest fuel types occupy most of the European territory (34.96 %), followed by cropland fuel types
(32.54 %). The fuel types with less representation in Europe are the wet and peat/semi-peat land (5.28 %) and the
shrubland (5.67 %) fuels. The only fuel type predominating in the arid/semi-arid regime is shrubland (75.45 %)
(Table B1 in Appendix B).

The application of the tested smoothing window sizes (3 x 3, 5 x 5, 7 x 7) increased the percentage of 10 x 10 pixel groups with unimodal distributions after resampling, although in all cases the increase was marginal (Table B2 in Appendix B). For all window sizes, more than 99 % of the pixel groups had a unimodal distribution, less than 1 % had a bimodal distribution, and only a few pixel groups presented a multimodal distribution of co-dominant categories. For the generation of the European fuel map at 1 km resolution, the 5 x 5 window was used as it provided a good compromise between generalisation and the level of detail preserved, maintaining important fuel types for fire behaviour typically made up of small clusters of pixels, such as urban discontinuous fabric.

The final European fuel map at 1 km resolution was generated, including 20 first-level fuel types (Fig. 5). The forest fuel types predominate in mountainous areas and the Scandinavian countries. The open and closed broadleaf deciduous forest, the open needleleaf evergreen forest, and the mixed forest are distributed over all Europe, while the closed needleleaf evergreen forest stands out in the Scandinavian region. The shrubland fuel types dominate in arid/semi-arid Europe. Most shrublands present medium and high depth. The grassland fuel types appear in cold areas (the Alps, the Scandinavian Mountains, the Pyrenees, etcetera) and are also important in Great Britain and Ireland, as rangelands. They are low in the arid/semi-arid region, medium in northern Europe, and high in central Europe. The herbaceous cropland fuel type is present all over Europe, while the woody cropland has lower importance, referring to fruit trees, vineyards, and olive trees in the Mediterranean area. The tree, shrubland and grassland wet and peat/semi-peat land fuel types occupy the Scandinavian Peninsula and northern Great Britain. Finally, the urban continuous fuel type relates to cities, and the urban discontinuous fuel type is distributed over all of Europe referring to the outskirts of cities and rural areas.

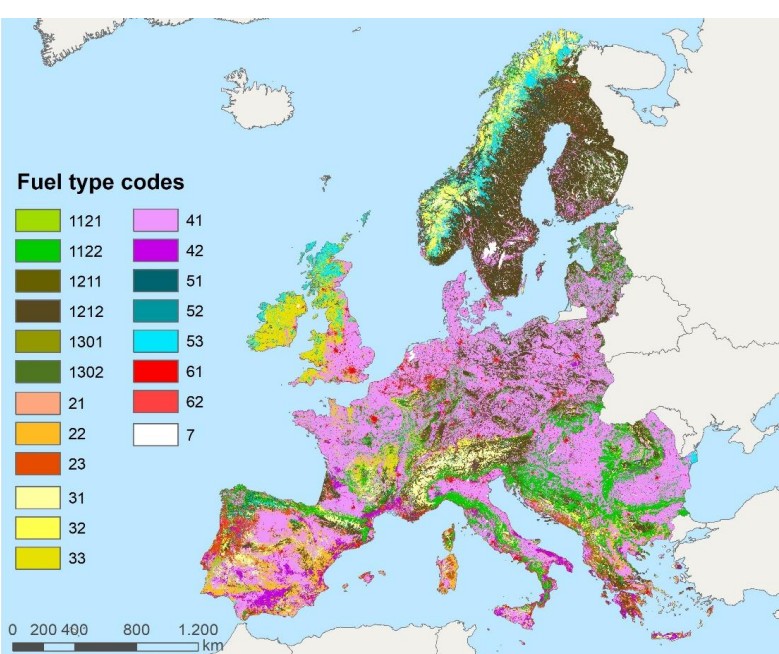

**Figure 5.** FirEUrisk European fuel map at 1 km resolution. See Table 1 for the fuel type codes identification.





In the final European fuel map at 1 km (Table 4), the fuel type dominating over Europe is cropland (38.52 %), mostly herbaceous (35.85 %), followed by the forest fuel types (32.66 %), mostly represented by the closed needleleaf evergreen forest (17.59 %). The fuel types with lower representation in Europe are urban (3.94 %) and wet and peat/semi-peat land (4.65 %). The only fuel types predominating in the arid/semi-arid regime are shrubland (> 83 %) and woody cropland (> 80 %).

**Table 4.** Area covered by every mapped FirEUrisk fuel type in Europe (1 km²). See Table 1 for the fuel type codes identification.

| FirEUrisk fuel type | Total area | | Area (%) by general bioclimatic regime | |
|---|---|---|---|---|
| | Thousands of km² | % | Arid/semi-arid | Sub-humid/humid |
| **Forest** | 1,606 | 32.66 | | |
| **1121** | 28 | 0.56 | 46.83 | 53.17 |
| **1122** | 454 | 9.23 | 15.81 | 84.19 |
| **1211** | 17 | 0.35 | 30.39 | 69.61 |
| **1212** | 865 | 17.59 | 6.58 | 93.42 |
| **1301** | 10 | 0.19 | 4.95 | 95.05 |
| **1302** | 233 | 4.74 | 3.92 | 96.08 |
| **Shrubland** | 265 | 5.39 | | |
| **21** | 6 | 0.12 | 99.88 | 0.12 |
| **22** | 140 | 2.84 | 88.44 | 11.56 |
| **23** | 119 | 2.43 | 83.12 | 16.88 |
| **Grassland** | 552 | 11.23 | | |
| **31** | 198 | 4.02 | 41.23 | 58.77 |
| **32** | 171 | 3.48 | 2.50 | 97.50 |
| **33** | 183 | 3.73 | 0.02 | 99.98 |
| **Cropland** | 1,894 | 38.52 | | |
| **41** | 1,763 | 35.85 | 18.61 | 81.39 |
| **42** | 131 | 2.67 | 80.52 | 19.48 |
| **Wet and peat/semi-peat land** | 229 | 4.65 | | |
| **51** | 57 | 1.16 | 9.30 | 90.70 |
| **52** | 6 | 0.13 | 35.52 | 64.48 |
| **53** | 165 | 3.36 | 4.66 | 95.34 |
| **Urban** | 194 | 3.94 | | |
| **61** | **100** | 2.03 | 18.64 | 81.36 |
| **62** | 94 | 1,91 | 20.17 | 79.83 |
| **Nonfuel** | 178 | 3.61 | 10.52 | 89.38 |

The validation of the European fuel map at 1 km resolution yielded a high overall agreement, 88.48 %, between the FirEUrisk European fuel map and the LUCAS points. Individual fuel types' accuracy ranged from 30



to 100 % (Table 5). As for the second validation exercise, including all mapped first-level FirEUrisk fuel types, a medium to a high quantitative agreement was observed (overall accuracy of 80.92 %). Individual fuel type's accuracy ranged from 20 to 100 % (Table 6, Table B3 in Appendix B).


**Table 5.** Confusion matrix for the FirEUrisk main fuel types. * UA: User accuracy (%), PA: Producer accuracy (%), CO: Commission error (%), OE: Omission error (%).

|  | Forest | Shr. | Grass. | Crop. | Wet. | Urban | Non. | Total | UA* | CE* |
|---|---|---|---|---|---|---|---|---|---|---|
| **Forest** | 1315 | 0 | 2 | 15 | 0 | 0 | 0 | 1332 | 98.72 | 1.28 |
| **Shr.** | 102 | 71 | 6 | 8 | 0 | 0 | 0 | 187 | 37.97 | 62.03 |
| **Grass.** | 14 | 20 | 196 | 17 | 2 | 0 | 0 | 249 | 78.71 | 21.29 |
| **Crop.** | 80 | 22 | 266 | 2838 | 3 | 0 | 2 | 3211 | 88.38 | 11.62 |
| **Wet.** | 2 | 6 | 3 | 0 | 6 | 0 | 0 | 17 | 35.29 | 64.71 |
| **Urban** | 1 | 0 | 0 | 0 | 0 | 9 | 0 | 10 | 90.00 | 10.00 |
| **Non.** | 1 | 2 | 3 | 0 | 1 | 0 | 3 | 10 | 30.00 | 70.00 |
| **Total** | 1515 | 121 | 476 | 2878 | 12 | 9 | 5 | 50016 |  |  |
| **PA*** | 86.80 | 58.68 | 41.18 | 98.61 | 50.00 | 100.00 | 60.00 | **Overall accuracy = 88.48 %** | | |
| **OE*** | 13.20 | 43.32 | 58.82 | 1.39 | 50.00 | 0.00 | 40.00 | | | |

**Table 6.** Accuracy summary for all mapped FirEUrisk fuel types. See Table 1 for the fuel type codes

identification. * CO: Commission error, OE: Omission error.

| FirEUrisk fuel type | CE (%)* | OE (%)* | FirEUrisk fuel type | CE (%)* | OE (%)* |
|---|---|---|---|---|---|
| **1121** | 70.00 | 70.00 | **32** | 40.00 | 80.00 |
| **1122** | 13.10 | 2.67 | **33** | 80.00 | 28.57 |
| **1211** | 20.00 | 72.41 | **41** | 7.77 | 0.76 |
| **1212** | 23.02 | 4.46 | **42** | 28.57 | 9.09 |
| **1301** | 30.00 | 56.25 | **51** | 80.00 | 50.00 |
| **1302** | 42.86 | 71.43 | **52** | 80.00 | 60.00 |
| **21** | 40.00 | 57.14 | **53** | 10.00 | 30.77 |
| **22** | 68.18 | 69.57 | **61** | 50.00 | 0.00 |
| **23** | 50.00 | 68.75 | **62** | 30.00 | 56.25 |
| **31** | 35.29 | 79.25 | **7** | 30.00 | 12.50 |
| **Overall accuracy = 80.92 %** | | | | | |

## 4 Fuel parameterization

### 4.1 Development of the crosswalk to standard fuel models

Once the fuel classification system was developed and used to map the European fuel types, we assigned

to each first-level FirEUrisk fuel type a surface fuel model: this allowed us to define surface fuel parameters at the continental scale. These parameters could be the input to run fire behaviour simulations, as well as for the estimation of fire risk conditions and fire effects. The main purpose of the crosswalk is to serve fire modelling



activities (e.g., spread and behaviour, emissions, post-fire, etcetera) because it allows mapping fuel models and their associated parameters.

The fuel types defined in this paper were matched to the Scott and Burgan Fire Behaviour Fuel Models (FBFM) (Scott and Burgan, 2005), which is a widely used fuel model classification system in Europe (Palaiologou et al., 2013; Aragoneses and Chuvieco, 2021; Alcasena et al., 2021). The FBFMs were based on the NFFL system (Anderson, 1982) and created to address fire behaviour predictions based on Rothermel's surface fire spread model (Rothermel, 1972) for the United States. They include 40 fuel models classified into 7 different groups according

to the predominant fire-carrying surface fuel type: grass (GR), grass-shrub (GS), shrub (SH), timber-understory (TU), timber-litter (TL), slash-blowdown (SB), and non-burnable (NB). Overall, the differences in fire behaviour among the surface fuel groups are mainly related to fuel load and its distribution among the particle size categories, Surface Area to Volume ratio, and fuel depth. Compared to NFFL models, the FBFM allows having a number of fuel models not fully cured or applicable in high-humidity areas. Regarding this point, to further improve the

matching possibility and account for variations in fuel types and moisture conditions across Europe, we distinguished arid/semi-arid and sub-humid/humid fuel types, as described in previous sections. Furthermore, FBFM data include more fuel models than the NFFL system for forest litter and litter with grass or shrub understory. Anyhow, a user can easily move from the proposed FBFMs to the NFFL system by using the crosswalk table between FBFM and NFFL fuel models (Scott and Burgan, 2005). In addition, our proposal of surface fuel

mapping and characterisation for the European general conditions can be adjusted or adapted to specific study areas or sites where more detailed information and measurements on fuels or custom data are available (Mutlu et al., 2008; Salis et al., 2016).

        For the purpose of this study, we assigned to each fuel type a given FBFM and the related fuel parameters that most fitted the average conditions in the field, according to expert knowledge. As a general rule, we assigned

grass models to fuel types related to grasslands and croplands and selected different sets of FBFM models depending on the fuel depth and cropland type, as well as on bioclimatic conditions: arid/semi-arid versus sub-humid/humid regimes (Fig. B1 in Appendix B). Shrub models were indicated in shrubland areas, following the same considerations described for grass models. Moreover, we proposed the use of shrub models in conditions of open forests, where the fractional cover is low, and the high availability of sunlight can stimulate the presence of

a shrubby understory. Timber understory and timber litter FBFMs were associated with closed forests: overall, we assigned low fuel-load models to evergreen forests and higher load models to broadleaf forests. The FirEUrisk fuel types 51, 52 and 53 were associated with shrub or grass FBFM models, depending on the main surface fuels. Finally, we proposed non-burnable (NB) conditions for urban continuous areas and other non-burnable zones (e.g., water, snow, ice, bare soils, sparse vegetation < 10 %), while shrub models were indicated for urban discontinuous

areas, to account for the potential of a fire to spread in such environments.

### 4.2 The FirEUrisk fuel classification system crosswalk to standard fuel models

        The FirEUrisk fuel types crosswalk to the FBFM system (Scott and Burgan, 2005) is presented in Table 7, and the related FBFM map over Europe is provided in Fig. 6 and complemented with Table 8.




**Table 7.** Suggested attribution of the first-level FirEUrisk fuel types to the FBFM standard fuel models in Europe. * A: arid/semi-arid regime, H: sub-humid/humid regime. See Table 1 for the fuel type codes identification and Table C1 in Appendix C for the FBFM descriptions and parameters.

| FirEUrisk fuel type | Crosswalk | | FirEUrisk fuel type | Crosswalk | |
|---|---|---|---|---|---|
| | A* | H* | | A* | H* |
| 1111 | SH7 | SH8 | 23 | SH5 | SH9 |
| 1112 | TU1 | TU2 | 31 | GR2 | GR6 |
| 1121 | SH5 | SH9 | 32 | GR4 | GR8 |
| 1122 | TU5 | TU3 | 33 | GR7 | GR9 |
| 1211 | SH7 | SH8 | 41 | GR4 | GR6 |
| 1212 | TU1 | TU2 | 42 | GR2 | GR6 |
| 1221 | SH5 | SH9 | 51 | SH7 | SH8 |
| 1222 | TU5 | TL3 | 52 | SH5 | SH9 |
| 1301 | SH7 | SH8 | 53 | GR7 | GR9 |
| 1302 | TU5 | TL3 | 61 | NB | NB |
| 21 | SH2 | SH3 | 62 | SH2 | SH3 |
| 22 | SH7 | SH8 | 7 | NB | NB |

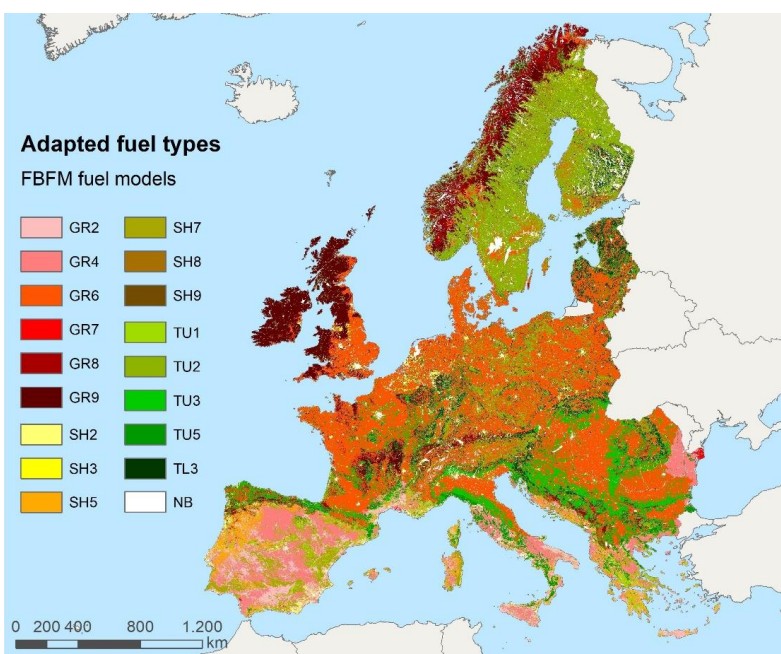


**Figure 6.** European fuel models based on the FBFM fuel models (Scott and Burgan, 2005) at 1 km resolution. See Table C1 in Appendix C for the fuel descriptions and parameters.



**Table 8.** Area covered by every FBFM fuel model in the European territory. See Table C1 in Appendix C for the fuel type descriptions and parameters.


| FBFM fuel model | Area | | FBFM fuel model | Area | |
|---|---|---|---|---|---|
| | Thousands of km² | % | | Thousands of km² | % |
| GR2 | 187 | 3.81 | SH7 | 134 | 2.73 |
| GR4 | 332 | 6.76 | SH8 | 89 | 1.81 |
| GR6 | 1,577 | 32.06 | SH9 | 39 | 0.79 |
| GR7 | 8 | 0.16 | TU1 | 57 | 1.16 |
| GR8 | 167 | 3.39 | TU2 | 808 | 16.44 |
| GR9 | 341 | 6.93 | TU3 | 382 | 7.77 |
| SH2 | 25 | 0.51 | TU5 | 81 | 1.65 |
| SH3 | 75 | 1.53 | TL3 | 224 | 4.55 |
| SH5 | 114 | 2.33 | NB | 277 | 5.64 |

The most relevant fuel model at the continental scale is GR6 (area covered about 1.6 Mkm²), which refers to medium-high and moderate live-load grasslands of sub-humid/humid areas and is characterised by high moisture values. This fuel model is largely related to herbaceous croplands that cover the most productive agricultural flat areas of central and northern Europe. About 0.8 Mkm² of Europe is covered by TU2, which was associated with

closed needleleaf evergreen forests located in the sub-humid/humid regime. TU2 is related to timber understory characterised by moderate-load shrubs. TU3, which concerns timber understory with a combined presence of grasses and shrubs with moderate fuel load, is the third more common fuel model in Europe, covering 7.77 % of the area. We proposed TU3 in closed broadleaf deciduous forests of sub-humid/humid areas. For arid/semi-arid

areas, GR4 is the dominant fuel model and occupies about 0.33 Mkm² (6.76 %) of land. This model represents moderate load grasses of dry climates. We associated GR4 with herbaceous croplands of southern Europe. Among the fuel models that cover more than 5 % of the study area, we should also mention the GR9, which refers to tall and high live load grasslands of sub-humid/humid areas and is characterised by high moisture values; and the non-burnable fuels, which refer to urban continuous areas and other non-burnable areas including bare soil, water, and

glaciers. The other FBFMs used in this work characterise approximately the remaining 24 % of the European territory and range from 0.22 Mkm² of TL3 to 7,721 km² of GR7.

A description of the parameters of the FBFM fuel models used for the crosswalk is presented in Table C1 in Appendix C. As an example, we mapped the 1h dead fuel load and the surface fuel depth over Europe (Fig. 7).

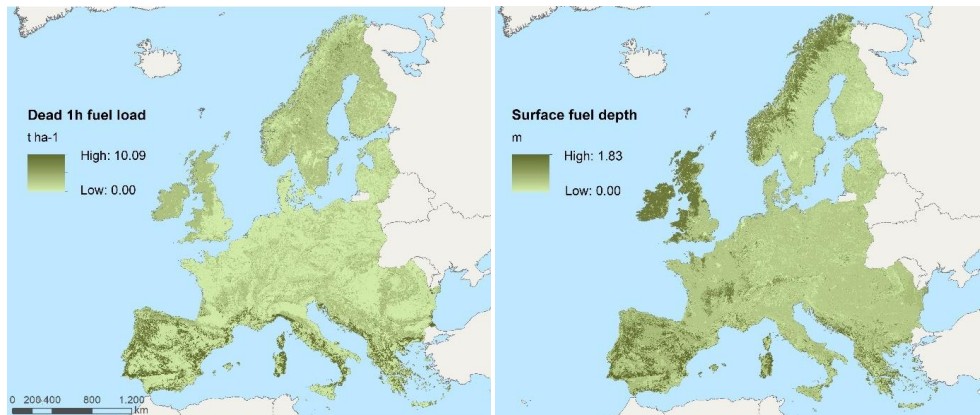


**Figure 7.** Surface dead 1h fuel load and fuel depth over Europe. Note that surface fuel depth for the forest fuels refers to the understory, not the crowns.

**5 Discussion**

The proposed FirEUrisk hierarchical fuel classification system was designed to be adapted to a wide range of environmental conditions, including those found in the European territory, describing both surface and canopy fuels. This constitutes an improvement in European fuel mapping compared to the global fuel map developed by Pettinari and Chuvieco (2016), which included more generic fuel categories, and the 2000 EFFIS fuel map (European Forest Fire Information System (EFFIS), 2017), only referring to surface fuels. The

hierarchical nature of the system aims to define a common fuel types' classification for all study areas and scales and offers high versatility because it is expected to enable fuel mapping at various scales with different disaggregation of categories, depending on the detail and quality of the input data. Thus, whereas the fuel map developed at the European scale was based on existing European and global datasets integrated into a GIS framework, the same classification scheme could be applied to provide a more comprehensive fuel classification

using a multi-sensor approach in a machine learning framework (García et al., 2011; Marino et al., 2016; Domingo et al., 2020). Its structure has similarities (e.g., hierarchical scheme) with the ArcFuel classification (Toukiloglou et al., 2013), although this was only prepared for southern-European conditions. In addition, the involvement of expert knowledge in the development of the FirEUrisk hierarchical fuel classification system suggests high acceptance, and therefore usage, among the fire risk management community in the foreseeable future. It also

allowed the development of a useful classification, intended to fill the actual gaps of the European fuel mapping, towards a homogeneous and integrated fire risk prevention strategy. Nevertheless, it must be considered that the grouping of vegetation types into fuel types is a balance between generalisation of the landscape reality and loss of detailed information, which may not be the most suitable system for all study areas.

        The predicted increase in fire intensity and occurrence of the so-called megafires (San-Miguel-Ayanz et

al., 2013), which usually evolve from surface to crown fires, makes it necessary to improve our information on canopy fuels. For this reason, our classification approach includes both surface and canopy fuels for the forest fuel types. The rest of the fuel types are disaggregated based on their fuel depth, with thresholds suggested by the experts. However, fuel mapping is still a challenge because of the high spatiotemporal variability of fuels, and the need to generalise the great variety of vegetation conditions related to fire behaviour.



Regarding the European fuel mapping, the combination of existing land cover and biogeographic datasets, and bioclimatic models, facilitated the generation of the fuel type dataset, being some of these data specifically adapted to European conditions (Europe's biogeographic regions map, CLC map). Nevertheless, the input datasets are a generalisation of the complex reality with their own uncertainties and errors, which are transferred to the final European fuel map. In fact, the errors of the final fuel type dataset are similar or even lower than those found

in the main input land cover map used to obtain the fuel categories.

Estimating shrubland and grassland fuel depth was challenging. To our knowledge, there are no large-scale reliable datasets in Europe on these variables. Although the models chosen to estimate fuel depth were not specifically developed for European areas, the biogeographical similarity to European conditions supported their usage for our purposes. For shrubland, 75 % of the shrubland fuels (in the 100 m resolution map) belong to the

arid/semi-arid regime, which justifies the selection of a bioclimatic model developed for an arid/semi-arid area. To avoid unrealistic estimations, we constrained the outputs to the range [0-6] m for the shrublands and to > 0 m for the grasslands, while no maximum cut-off threshold was applied to the grassland category as the obtained maximum value (1 m) was considered reasonable. In addition, the distribution of shrubland and grassland pixels led to considering the bioclimatic models adequate. The histogram for shrubland fuel depth showed the spatial

continuity of the input variable (precipitation). The histogram for grassland fuel depth had an aggregated structure due to the input productivity data by biogeographic region. Obviously, direct measurement of shrubland or grassland fuel depth is more desirable. Future works based on airborne or satellite LiDAR should provide a better estimation, but they are not yet available for the whole European territory (airborne) or need addition calibration efforts (satellite).

Concerning the final European fuel map (1 km$^2$), only 20 out of the 24 possible first-level fuel types were mapped because for a fuel type to be mapped, it must occupy a continuous area large enough to be represented in 1 km$^2$. The herbaceous cropland and the closed needleleaf evergreen forests are the most extended fuel types in Europe, related to the land use activities of the European society and the natural distribution of vegetation species due to bioclimatic conditions (García-Martín et al., 2001). Also, the large extension of forest fuel types constitutes

an increasing potential risk in the light of the growing trends of land abandonment, particularly in remote areas: forests with high surface fuel load can more easily turn into crown fires (Scott and Reinhardt, 2001; Weise and Wright, 2014), characterised by high intensity, emitting vast amounts of the stored carbon. Urban fuel types are the least represented in Europe, but they are the most dangerous from an economic, societal and human health point of view (Bowman et al., 2011).

Finally, the quantitative assessment of the mapped FirEUrisk main fuel types (forest, shrubland, grassland, cropland, wet and peat/semi-peat land, and urban), plus the nonfuel category; obtained a high overall accuracy of 88.48 %: average commission errors of 34 % (highest for the nonfuel category and lowest for the forest fuel types) and average omission errors 30 % (highest for the grassland and lowest for the urban fuel types). Although it is higher than the used base cartography, the Copernicus GLC map (Tsendbazar et al., 2020), and it

surpassed the ideal 85 % minimum overall accuracy; not all fuel types presented the ideal ≥ 70 % accuracy (Thomlinson et al., 1999). The overall accuracy was higher than the one for the 2019 Copernicus GLC map over Europe (79.9 %), probably due to the validation approach. The confusion matrix is aligned with the confusion matrix of the 2015 global Copernicus GLC maps over Europe (Tsendbazar et al., 2020), considering most similar



categories. The errors of the Copernicus GLC map have been transferred to the European fuel map as it was used

as base cartography.

      With similar accuracies as the 2015 Copernicus GLC map over Europe, forest fuel types present low omission and commission errors, although there is some confusion with shrubland, grassland, and cropland. The shrubland omission and commission errors (mostly confused by the Mediterranean sclerophyllous and xerophilic forest) are significant, however, our validation approach obtained 16 % and 2 % less, respectively, compared to

the 2015 Copernicus GLC map. The grassland omission errors (mostly confused by herbaceous cropland) are 15 % higher than the ones for herbaceous vegetation in the 2015 Copernicus GLC map. In addition, grassland commission errors are 16 % lower than in the 2015 Copernicus GLC map. Croplands present higher (+7 %) producer and user accuracies than the 2015 Copernicus GLC map, mostly confused with grassland, being the producer accuracy higher than the user accuracy as in the Copernicus GLC map. Wet and peat/semi-peat land

omission errors are 3 % lower and commission errors are 11 % higher than in the 2015 Copernicus GLC map for herbaceous wetland, in agreement with the observed accuracy tendencies (Tsendbazar et al., 2021). Urban fuel types have the lowest omission error (0 %), and only 10 % of commission error. The nonfuel category errors are mostly referred to pixels over the coastline caused by the different spatial resolutions of the European fuel map and the LUCAS points. This also happens to the rest of the fuel types and is considered the main limitation of the

validation method. Some validation errors are also caused by the different dates of the input sources and the validation data.

      The quantitative assessment of all mapped FirEUrisk fuel types obtained a medium-high overall accuracy of 80.92 %: average commission errors of 41 % (highest for the high grasslands, and tree and shrubland wet and peat/semi-peat land fuel types; and lowest for the herbaceous cropland fuel type) and average omission errors of

50 % (highest for the medium grassland fuel type and lowest for the urban continuous fabric fuel type). These results are higher than the used base cartography, the Copernicus GLC map (Tsendbazar et al., 2020), but do not surpass the ideal 85 % minimum overall accuracy, neither all fuel types with ≥ 70 % accuracy (Thomlinson et al., 1999). However, the visual assessment improved the validation method because it considered the entire 1 km$^2$ pixels and not only the area of the LUCAS points. This method could only be applied to a subset of the validation

points because of its temporal and human cost compared to the previous validation method. The results are similar to the confusion matrices of the FirEUrisk main fuel types and the Copernicus GLC map over Europe (Tsendbazar et al., 2020), although errors are higher and different due to the dissimilar validation methods and reference data, and that confusion appears between fuel types belonging to the same main fuel type. Most errors are due to pixels with a mixed cover of fuel types, and low quality of the reference data (unclear and blurred Google images and

LUCAS photos; and pixels not meeting all ideal conditions for validation - that was needed to have a representative sampling for every fuel type). Input and reference data temporal differences can also have affected the accuracy. The obtained errors present the typical pattern for land cover and vegetation classifications with remote sensing (used to develop the input data), dependent on the separability of the spectral signatures of the land types. This explains why errors are dominant for fuel types belonging to the same main fuel type instead of fuel types from

different main fuel types.

      Forest fuel types have acceptable accuracy except for the closed mixed forest, highly confused with closed needleleaf evergreen forest. Many errors refer to the omission of open forest, assigned to the closed forest, as happens in the Copernicus GLC map over Europe (Tsendbazar et al., 2020). Shrubland and grassland fuel types'





errors are significant, mostly between fuel depth categories. However, care must be taken for these results, as estimating fuel depth from photos is challenging, and fuel depth varies with time. These limitations specially affect grassland due to its low depth, rapid growth, and that high grassland is frequently cut. Thus, grassland fuel depth is very changeable so we assume the European fuel map may only be accurate for some periods of the year. We validated the proposed fuel map considering the mean potential fuel depth. Moreover, short grassland is generally confused with herbaceous cropland of fodder crops of agriculturally improved grasslands and temporary pasture such as legumes. Cropland fuel types are the most accurate, with no significant errors. Wet and peat/semi-peat land fuel types have moderate accuracy. It outstands the confusion of tree wet and peat/semi-peat land with other wet and peat/semi-peat land fuel types, and shrubland wet and peat/semi-peat land with shrubland. The urban continuous fuel type has no omission errors, while some commission errors are in favour of the urban discontinuous fuel type in the outskirt's residential areas of cities. The urban discontinuous fuel type presents higher omission than commission errors, mostly omitted by cropland in agricultural rural areas. Similar to the confusion matrix for the main fuel types, both commission and omission errors for the nonfuel category are low (≤ 30 %) and relate to mixed pixels.

The different levels of disaggregation of the proposed classification system, as well as the main fire behaviour characteristics of the diverse fuels, made the crosswalk challenging and did not allow to assign a specific standard model to each FirEUrisk fuel type. Moreover, the FBFM standard fuel models (Scott and Burgan, 2005) were originally developed for the United States, so care must be taken when using the crosswalk in Europe (Santoni et al., 2011; Salis et al., 2016). From this point of view, our proposed approach can be improved in specific areas if customised information and data on given fuel types are available (Arca et al., 2007; Fernandes, 2009; Duguy Pedra et al., 2015; Kucuk et al., 2015; Ascoli et al., 2020). In other words, we propose a generic crosswalk scheme, but users are free to wisely choose or modify the best fitting standard fuel models according to their study area and expertise, or to use different parameters from the standard ones if they have better information for given study areas. Plus, the main limitation of the crosswalk scheme relies on the reference to general bioclimatic regimes, which is not able to fully consider all inherent differences among European regions in terms of fuel characteristics, while moisture values can be spatially modified according to the specific status of each fuel type.

This work represents one of the first attempts to adopt a standardised fuel model mapping approach over Europe, similar to the National Fire Danger Rating fuel models products available since the '90s for the continental United States (see for instance https://www.wfas.net/index.php/nfdrs-fuel-model-static-maps-44). Work is in progress to develop higher resolution products over Europe combining a set of remote sensing tools and data. This latter development at the European scale is highly complicated by the huge heterogeneity in the availability of high quality and resolution of ground and measured data, which vary a lot among and within the different regions.

The FirEUrisk fuel classification system can provide a number of insights and information for wildfire risk monitoring and assessment at the European scale. This is mostly related to the identified fuel categories crosswalk to the FBFM system (Scott and Burgan, 2005), which is specifically designed for the above purpose. In fact, the parameters included in each FBFM model allow the characterization of surface fuels and can serve as a baseline for surface wildfire spread and behaviour modelling. For instance, some existing fire spread models, such as FlamMap (Finney, 2006) and FARSITE (Finney, 2004), could be used for this purpose, although canopy parameters should be additionally estimated. This should be subject of an extension of this project and could be based on airborne and satellite LiDAR systems.



The fuel map is also expected to serve estimations of fire-caused carbon emissions and pollution, and estimations of biomass consumption. The Consume model (Prichard et al., 2006) could be used for this if a crosswalk to FCCS fuels is previously made, including the necessary fuel parameters such as the combustion percentage. In addition, the European fuel map would be useful for regions that do not have fuel cartography.

Overall, we highlight that the main use of the map is providing a dataset able to rate fire danger and risk conditions across large geographic areas, while the application of wildfire spread models to very local scales or small areas may pose limitations in the quality of outputs due to low resolution (1 km$^2$) of the fuel input layer.

Finally, although it has been developed for European conditions, our methodology has the potential to be applied to other regions. The proposed fuel classification system could be used in other projects and applications apart from the FirEUrisk project, and adapted anywhere in the world, further extending the fuel subcategories wherever required. The classification of fuel types is dependent on existing land cover and biogeographic data, but it can also be directly estimated from satellite data, either coarse resolution for continental areas or higher resolution for smaller territories. The fuel parameterization can also be based on standard models, such as the Scott and Burgan system (Scott and Burgan, 2005) used in this paper, but it can also rely on ground measurements or more detailed regional fuel characteristics. In any case, it is important to emphasise the need of estimating fuel parameters to use the fuel type products for quantitative estimations of fire risk, behaviour, and effects. This is a key aspect of the FirEUrisk project and a crucial point toward wildland fire prevention across the European Union.

**6 Data availability**

The resulting European fuel map (1 km$^2$) in one single-band categorical raster layer in GeoTIFF format is publicly available at https://doi.org/10.21950/YABYCN (Aragoneses et al., 2022a), as well as a Product User Manual (PUM) (Aragoneses et al., 2022b), at *e-cienciaDatos:* https://edatos.consorciomadrono.es/dataset.xhtml?persistentId=doi:10.21950/YABYCN.

**7 Conclusions**

This work, developed in the framework of the European FirEUrisk project , provided a hierarchical fuel classification system for surface and canopy fuels adapted to continental conditions. The final European fuel map contains 20 fuel types, including both surface and canopy fuel categories. The estimated overall accuracy was 88 % for the main fuel types and 81 % for all mapped fuel types. A crosswalk between the proposed fuel types and commonly used standard fuel models, Fire Behaviour Fuel Models (FBFM) (Scott and Burgan, 2005), has been presented as well. Our approach, based on expert knowledge, Geographic Information Systems, existing land cover datasets, biogeographic data, and bioclimatic modelling, could be readily applied to other regions.

The results of this study constitute the first step toward a risk-wise landscape and fuel mapping development across Europe, which will help integrated, strategic, coherent, and comprehensive decision making for fire risk prevention, assessment, and evaluation. The results have wide applicability because they meet the actual unfulfilled fuel mapping needs in Europe, allowing to coordinate fuel mapping at different scales and across European regions.



## Appendix A

Table A1: The FirEUrisk hierarchical fuel classification system.

| First-level | | | | Second-level | |
|---|---|---|---|---|---|
| **Main fuel types** | **Leaf type/ Type** | **Phenology** | **Fractional cover (%)** | **Understory type** | **Understory depth** |
| 1. Forest | 11. Broadleaf | 111. Evergreen | 1111. Open [15-70 %) | 3. Grassland | 31. Low [0-0.3 m) |
| | | | | | 32. Medium [0.3-0.7 m) |
| | | | | | 33. High (≥ 0.7 m) |
| | | | | 2. Shrubland | 21. Low [0-0.5 m) |
| | | | | | 22. Medium [0.5-1.5 m) |
| | | | | | 23. High (≥ 1.5 m) |
| | | | | 0. Timber litter | |
| | | | 1112. Closed [70-100 %) | 3. Grassland | 31. Low [0-0.3 m) |
| | | | | | 32. Medium [0.3-0.7 m) |
| | | | | | 33. High (≥ 0.7 m) |
| | | | | 2. Shrubland | 21. Low [0-0.5 m) |
| | | | | | 22. Medium [0.5-1.5 m) |
| | | | | | 23. High (≥ 1.5 m) |
| | | | | 0. Timber litter | |
| | | 112. Deciduous | 1121. Open [15-70 %) | 3. Grassland | 31. Low [0-0.3 m) |
| | | | | | 32. Medium [0.3-0.7 m) |
| | | | | | 33. High (≥ 0.7 m) |
| | | | | 2. Shrubland | 21. Low [0-0.5 m) |
| | | | | | 22. Medium [0.5-1.5 m) |
| | | | | | 23. High (≥ 1.5 m) |
| | | | | 0. Timber litter | |
| | | | 1122. Closed [70-100 %) | 3. Grassland | 31. Low [0-0.3 m) |
| | | | | | 32. Medium [0.3-0.7 m) |
| | | | | | 33. High (≥ 0.7 m) |
| | | | | 2. Shrubland | 21. Low [0-0.5 m) |
| | | | | | 22. Medium [0.5-1.5 m) |
| | | | | | 23. High (≥ 1.5 m) |
| | | | | 0. Timber litter | |
| | 12. Needleleaf | 121. Evergreen | 1211. Open [15-70 %) | 3. Grassland | 31. Low [0-0.3 m) |
| | | | | | 32. Medium [0.3-0.7 m) |
| | | | | | 33. High (≥ 0.7 m) |
| | | | | 2. Shrubland | 21. Low [0-0.5 m) |
| | | | | | 22. Medium [0.5-1.5 m) |
| | | | | | 23. High (≥ 1.5 m) |



| | | | | |
|---|---|---|---|---|
| | | | 0. Timber litter | |
| | | 1212. Closed [70-100 %) | 3. Grassland | 31. Low [0-0.3 m) |
| | | | | 32. Medium [0.3-0.7 m) |
| | | | | 33. High (≥ 0.7 m) |
| | | | 2. Shrubland | 21. Low [0-0.5 m) |
| | | | | 22. Medium [0.5-1.5 m) |
| | | | | 23. High (≥ 1.5 m) |
| | | | 0. Timber litter | |
| | 122. Deciduous | 1221. Open [15-70 %) | 3. Grassland | 31. Low [0-0.3 m) |
| | | | | 32. Medium [0.3-0.7 m) |
| | | | | 33. High (≥ 0.7 m) |
| | | | 2. Shrubland | 21. Low [0-0.5 m) |
| | | | | 22. Medium [0.5-1.5 m) |
| | | | | 23. High (≥ 1.5 m) |
| | | | 0. Timber litter | |
| | | 1222. Closed [70-100 %) | 3. Grassland | 31. Low [0-0.3 m) |
| | | | | 32. Medium [0.3-0.7 m) |
| | | | | 33. High (≥ 0.7 m) |
| | | | 2. Shrubland | 21. Low [0-0.5 m) |
| | | | | 22. Medium [0.5-1.5 m) |
| | | | | 23. High (≥ 1.5 m) |
| | | | 0. Timber litter | |
| | 13. Mixed | 1301. Open [15-70 %) | 3. Grassland | 31. Low [0-0.3 m) |
| | | | | 32. Medium [0.3-0.7 m) |
| | | | | 33. High (≥ 0.7 m) |
| | | | 2. Shrubland | 21. Low [0-0.5 m) |
| | | | | 22. Medium [0.5-1.5 m) |
| | | | | 23. High (≥ 1.5 m) |
| | | | 0. Timber litter | |
| | | 1302. Closed [70-100 %) | 3. Grassland | 31. Low [0-0.3 m) |
| | | | | 32. Medium [0.3-0.7 m) |
| | | | | 33. High (≥ 0.7 m) |
| | | | 2. Shrubland | 21. Low [0-0.5 m) |
| | | | | 22. Medium [0.5-1.5 m) |
| | | | | 23. High (≥ 1.5 m) |
| | | | 0. Timber litter | |
| | **Fuel depth** | | | |
| 2. Shrubland | 21. Low [0-0.5 m) | | | |
| | 22. Medium [0.5-1.5 m) | | | |
| | 23. High (≥ 1.5 m) | | | |
| 3. Grassland | 31. Low [0-0.3 m) | | | |
| | 32. Medium [0.3-0.7 m) | | | |



| | 33. High (≥ 0.7 m) | |
|---|---|---|
| | **Type** | |
| 4. Cropland | 41. Herbaceous | |
| | 42. Woody (shrub-tree) | |
| 5. Wet and peat/ semi-peat land | 51. Tree | |
| | 52. Shrubland | |
| | 53. Grassland | |
| 6. Urban | 61. Continuous fabric: urban fabric (≥ 80 %) | |
| | 62. Discontinuous fabric: vegetation and urban fabric [15-80 %] | |
| 7. Nonfuel | | 71. Water/snow/ice |
| | | 72. Bare soil/sparse vegetation (< 10 %) |

**Appendix B**

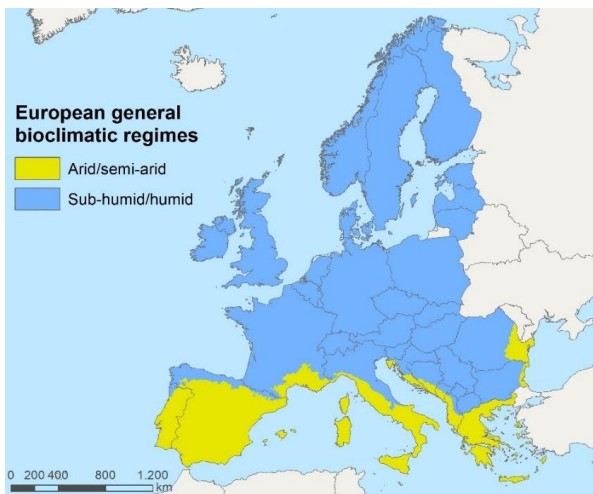

**Figure B1.** Location of the arid/semi-arid and sub-humid/humid regimes over Europe.

**Table B1.** Area covered by every FirEUrisk main fuel type at 100 m resolution in Europe.

| FirEUrsik main fuel type | Total area | | Area (%) by general bioclimatic regime | |
|---|---|---|---|---|
| | **Thousands of km²** | **%** | **Arid/semi-arid** | **Sub-humid/humid** |
| **Forest** | 17 | 34.96 | 10.84 | 89.16 |
| **Shrubland** | 3 | 5.67 | 75.45 | 24.55 |
| **Grassland** | 5 | 10.71 | 16.62 | 83.38 |
| **Cropland** | 16 | 32.54 | 24.63 | 75.37 |
| **Wet and peat/semi-peat land** | 3 | 5.28 | 6.65 | 93.35 |
| **Urban** | 4 | 7.26 | 17.70 | 82.30 |
| **Nonfuel** | 2 | 3.58 | 7.65 | 92.35 |





**Table B2.** Percentage of 10 x 10 pixel groups with 1, 2 or > 2 first-mode categories for the 3 x 3, 5 x 5, and 7 x 7 smoothing moving windows, and without window applied.

| Window size | Percentage (%) of 10 x 10 pixel groups with: | | |
|---|---|---|---|
| | 1 first-mode category | 2 first-mode categories | > 2 first-mode categories |
| **No window** | 99.27 | 0.72 | 0.01 |
| **3 x 3** | 99.39 | 0.60 | 0.01 |
| **5 x 5** | 99.48 | 0.51 | 0 |
| **7 x 7** | 99.54 | 0.45 | 0 |



**Table B3.** Confusion matrix for all mapped FirEUrisk fuel types. See Table 1 for the fuel type codes identification.

* T: Total, UA: User accuracy (%), PA: Producer accuracy (%), CO: Commission error (%), OE: Omission error (%).

| | 1121 | 1122 | 1211 | 1212 | 1301 | 1302 | 21 | 22 | 23 | 31 | 32 | 33 | 41 | 42 | 51 | 52 | 53 | 61 | 62 | 7 | T* | UA* | CE* |
|---|---|---|---|---|---|---|---|---|---|---|---|---|---|---|---|---|---|---|---|---|---|---|---|
| 112 1 | 3 | 1 | 1 | 0 | 0 | 0 | 0 | 1 | 0 | 3 | 1 | 0 | 0 | 0 | 0 | 0 | 0 | 0 | 0 | 0 | 10 | 30.00 | 70.00 |
| 112 2 | 5 | 73 | 0 | 0 | 1 | 2 | 0 | 0 | 1 | 0 | 0 | 0 | 1 | 0 | 0 | 0 | 0 | 0 | 1 | 0 | 84 | 86.90 | 13.10 |
| 121 1 | 0 | 0 | 8 | 0 | 0 | 0 | 0 | 0 | 1 | 0 | 0 | 0 | 0 | 0 | 1 | 0 | 0 | 0 | 0 | 0 | 10 | 80.00 | 20.00 |
| 121 2 | 0 | 0 | 12 | 107 | 0 | 17 | 0 | 0 | 0 | 1 | 2 | 0 | 0 | 0 | 0 | 0 | 0 | 0 | 0 | 0 | 139 | 76.98 | 23.02 |
| 130 1 | 0 | 0 | 0 | 2 | 7 | 0 | 0 | 0 | 0 | 0 | 1 | 0 | 0 | 0 | 0 | 0 | 0 | 0 | 0 | 0 | 10 | 70.00 | 30.00 |
| 130 2 | 0 | 1 | 0 | 1 | 4 | 8 | 0 | 0 | 0 | 0 | 0 | 0 | 0 | 0 | 0 | 0 | 0 | 0 | 0 | 0 | 14 | 57.14 | 42.86 |
| 21 | 0 | 0 | 1 | 0 | 0 | 0 | 6 | 2 | 1 | 0 | 0 | 0 | 0 | 0 | 0 | 0 | 0 | 0 | 0 | 0 | 10 | 60.00 | 40.00 |
| 22 | 1 | 0 | 2 | 0 | 1 | 0 | 4 | 7 | 7 | 0 | 0 | 0 | 0 | 0 | 0 | 0 | 0 | 0 | 0 | 0 | 22 | 31.82 | 68.18 |
| 23 | 0 | 0 | 0 | 0 | 0 | 0 | 0 | 3 | 5 | 2 | 0 | 0 | 0 | 0 | 0 | 0 | 0 | 0 | 0 | 0 | 10 | 50.00 | 50.00 |
| 31 | 0 | 0 | 0 | 0 | 0 | 0 | 1 | 1 | 0 | 11 | 3 | 0 | 0 | 0 | 0 | 0 | 0 | 0 | 0 | 1 | 17 | 64.71 | 35.29 |
| 32 | 0 | 0 | 0 | 0 | 0 | 0 | 1 | 1 | 0 | 2 | 6 | 0 | 0 | 0 | 0 | 0 | 0 | 0 | 0 | 0 | 10 | 60.00 | 40.00 |
| 33 | 0 | 0 | 0 | 0 | 0 | 0 | 0 | 0 | 0 | 13 | 7 | 5 | 0 | 0 | 0 | 0 | 0 | 0 | 0 | 0 | 25 | 20.00 | 80.00 |
| 41 | 0 | 0 | 3 | 0 | 2 | 1 | 0 | 1 | 0 | 21 | 10 | 1 | 522 | 1 | 0 | 0 | 0 | 0 | 4 | 0 | 566 | 92.23 | 7.77 |
| 42 | 0 | 0 | 0 | 0 | 1 | 0 | 0 | 0 | 1 | 0 | 0 | 0 | 2 | 10 | 0 | 0 | 0 | 0 | 0 | 0 | 14 | 71.43 | 28.57 |
| 51 | 1 | 0 | 2 | 0 | 0 | 0 | 0 | 1 | 0 | 0 | 0 | 0 | 0 | 0 | 2 | 2 | 2 | 0 | 0 | 0 | 10 | 20.00 | 80.00 |
| 52 | 0 | 0 | 0 | 0 | 0 | 0 | 2 | 6 | 0 | 0 | 0 | 0 | 0 | 0 | 0 | 2 | 0 | 0 | 0 | 0 | 10 | 20.00 | 80.00 |
| 53 | 0 | 0 | 0 | 0 | 0 | 0 | 0 | 0 | 0 | 0 | 0 | 0 | 0 | 0 | 0 | 1 | 9 | 0 | 0 | 0 | 10 | 90.00 | 10.00 |
| 61 | 0 | 0 | 0 | 1 | 0 | 0 | 0 | 0 | 0 | 0 | 0 | 0 | 0 | 0 | 0 | 0 | 0 | 5 | 4 | 0 | 10 | 50.00 | 50.00 |
| 62 | 0 | 0 | 0 | 0 | 0 | 0 | 0 | 0 | 0 | 0 | 0 | 0 | 1 | 0 | 1 | 0 | 1 | 0 | 7 | 0 | 10 | 70.00 | 30.00 |
| 7 | 0 | 0 | 0 | 1 | 0 | 0 | 0 | 0 | 0 | 0 | 0 | 1 | 0 | 0 | 0 | 0 | 1 | 0 | 0 | 7 | 10 | 70.00 | 30.00 |
| T* | 10 | 75 | 29 | 112 | 16 | 28 | 14 | 23 | 16 | 53 | 30 | 7 | 526 | 11 | 4 | 5 | 13 | 5 | 16 | 8 | 1001 | | |
| PA* | 30.00 | 97.33 | 27.59 | 95.54 | 43.75 | 28.57 | 42.86 | 30.43 | 31.25 | 20.75 | 20.00 | 71.43 | 99.24 | 90.91 | 50.00 | 40.00 | 69.23 | 100.00 | 43.75 | 87.50 | | **Overall accuracy = 80.92 %** | |
| OE* | 70.00 | 2.67 | 72.41 | 4.46 | 56.25 | 71.43 | 57.14 | 69.57 | 68.75 | 79.25 | 80.00 | 28.57 | 0.76 | 9.09 | 50.00 | 60.00 | 30.77 | 0.00 | 56.25 | 12.50 | | | |


**Appendix C**

**Table C1.** Parameters of the standard fuel models of FBFM (Scott and Burgan, 2005) used for the crosswalk to the first-level FirEUrisk fuel types.

| FBFM fuel model | Dead fuel load (t ha⁻¹) | | | Live fuel load (t ha⁻¹) | | Surface Area to Volume ratio (m² m⁻³) | | | Depth (m) | Moisture of extinction (%) | Heat content (kj kg⁻¹) | | Main fuel type | Description |
|---|---|---|---|---|---|---|---|---|---|---|---|---|---|---|
| | 1h | 10h | 100h | Herb | Woody | Dead 1h | Live herb | Live woody | | | Dead | Live | | |
| GR2 | 0.22 | 0.00 | 0.00 | 2.24 | 0.00 | 6562 | 5906 | 4921 | 0.30 | 15 | 18622 | 18622 | Grasses | Low load. Dry climate grass |
| GR4 | 0.56 | 0.00 | 0.00 | 4.26 | 0.00 | 6562 | 5906 | 4921 | 0.61 | 15 | 18622 | 18622 | Grasses | Moderate load. Dry climate grass |
| GR6 | 0.22 | 0.00 | 0.00 | 7.62 | 0.00 | 7218 | 6562 | 4921 | 0.46 | 40 | 18622 | 18622 | Grasses | Moderate load. Humid climate grass |
| GR7 | 2.24 | 0.00 | 0.00 | 12.11 | 0.00 | 6562 | 5906 | 4921 | 0.91 | 15 | 18622 | 18622 | Grasses | High load. Dry climate grass |
| GR8 | 1.12 | 2.24 | 0.00 | 16.36 | 0.00 | 4921 | 4265 | 4921 | 1.22 | 30 | 18622 | 18622 | Grasses | High load. Very coarse. Humid climate grass |
| GR9 | 2.24 | 2.24 | 0.00 | 20.18 | 0.00 | 5906 | 5249 | 4921 | 1.52 | 40 | 18622 | 18622 | Grasses | Very high load. Humid climate grass |
| SH2 | 3.03 | 5.38 | 1.68 | 0.00 | 8.63 | 6562 | 4921 | 5249 | 0.30 | 15 | 18622 | 18622 | Shrubs | Moderate load. Dry climate shrub |
| SH3 | 1.01 | 6.73 | 0.00 | 0.00 | 13.90 | 5249 | 4921 | 4593 | 0.73 | 40 | 18622 | 18622 | Shrubs | Moderate load. Humid climate shrub |
| SH5 | 8.07 | 4.71 | 0.00 | 0.00 | 6.50 | 2461 | 4921 | 5249 | 1.83 | 15 | 18622 | 18622 | Shrubs | High load. Dry climate shrub |
| SH7 | 7.85 | 11.88 | 4.93 | 0.00 | 7.62 | 2461 | 4921 | 5249 | 1.83 | 15 | 18622 | 18622 | Shrubs | Remarkably high load. Dry climate shrub |
| SH8 | 4.60 | 7.62 | 1.91 | 0.00 | 9.75 | 2461 | 4921 | 5249 | 0.91 | 40 | 18622 | 18622 | Shrubs | High load. Humid climate shrub |
| SH9 | 10.09 | 5.49 | 0.00 | 3.47 | 15.69 | 2461 | 5906 | 4921 | 1.34 | 40 | 18622 | 18622 | Shrubs | Remarkably high load. Humid climate shrub |
| TU1 | 0.45 | 2.02 | 3.36 | 0.45 | 2.02 | 6562 | 5906 | 5249 | 0.18 | 20 | 18622 | 18622 | Litter & Understory | Low load. Dry climate timber-grass-shrub |
| TU2 | 2.13 | 4.04 | 2.80 | 0.00 | 0.45 | 6562 | 4921 | 5249 | 0.30 | 30 | 18622 | 18622 | Litter & Understory | Moderate load. Humid climate timber-shrub |
| TU3 | 2.47 | 0.34 | 0.56 | 1.46 | 2.47 | 5906 | 5249 | 4593 | 0.40 | 30 | 18622 | 18622 | Litter & Understory | Moderate load. Humid climate timber-grass-shrub |
| TU5 | 8.97 | 8.97 | 6.73 | 0.00 | 6.73 | 4921 | 4921 | 2461 | 0.30 | 25 | 18622 | 18622 | Litter & Understory | Very high load. Dry climate timber-shrub |
| TL3 | 1.12 | 4.93 | 6.28 | 0.00 | 0.00 | 6562 | 4921 | 4921 | 0.09 | 20 | 18622 | 18622 | Litter & Understory | Moderate load conifer litter |



**Author contributions.** Conceptualization, writing—review and editing E.A., M.G., M.S., LM.R. and E.C.; methodology, resources, E.A., M.G. and E.C.; data curation, formal analysis, investigation, software, validation, visualisation, writing—original draft E.A.; supervision, M.G. and E.C.; project administration, funding acquisition, E.C. All authors have read and agreed to the published version of the manuscript.

**Competing interests.** The authors declare no conflict of interest.

**Disclaimer.** This research reflects only the authors' view, and the European Commission is not responsible for any use that may be made of the information it contains.

**Acknowledgements.** We would like to thank the members of the FirEUrisk consortium for their comments on the FirEUrisk fuel classification system. We would also like to thank María Clara Ochoa and Suresh Babu Kukkala for their help with the product's validation. We are also grateful to anonymous reviewers for their helpful comments.

**Financial support.** This project has been granted funding from the European Union's Horizon 2020 research and innovation programme under Grant Agreement No. 101003890.

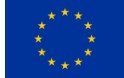

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
