# Peer review of "Classification and mapping of European fuels using a hierarchical-multipurpose fuel classification system"

_Earth System Science Data, 2022_

## Referee Comment (RC2)

[referee-annotated manuscript omitted]

---

## Author Comment (AC1)

**Response to Anonymous Reviewer #1**

Dear authors,

This work provides a hierarchical classification system of surface and canopy fuels adapted to continental conditions, covering 85 fuel types, and grouped into six major fuel categories. The final European fuel map contains 20 fuel types, obtaining a high estimated overall accuracy (88%-81%) for all mapped fuel types.

I reaffirm the words of the authors and it is the first step towards a mapping that takes into account the risk of risk throughout Europe, which can help in decision making from a prevention point of view that in years like 2022 in the Mediterranean regions a pre-season fire risk assessment is necessary.

It only remains for me to congratulate the authors since the article brings a novelty in the state of the art, it has great quality both in the content and in the presentation of the same.

To mention it, I would only detail the processing time required and the work process followed.

Response: The authors thank the reviewer for taking the time to read the manuscript and appreciate the congratulations on the work done.

Attending to the reviewer suggestion, we have detailed the process followed by modifying Figure 1 as follows. Regarding the required processing time, once the methodology is prepared, the generation of the European fuel map can be done in some days (although validation efforts would require more time).

[Figure]

Figure 1. General overview of the structure of this work.

---

## Author Comment (AC2)

**Response to Anonymous Reviewer #2**

In this paper the authors propose a new fuel classification system for Europe and release a fuel map at 1 km spatial resolution. I accepted to review this paper because of the title and the abstract promise to fill one of the main gap in forest fire research. However, I read this paper several times, but it is really difficult to understand how this work fills the gap of the lack of appropriate fuel mapping in Europe.

Response: The authors thank the reviewer for taking the time to review our manuscript. We also appreciate very much the advises and suggestions made. We have thoroughly considered the suggestions made. Improvements have been made in order to further explain and clarify the novelty of the work and how it tries to fill the actual gaps of the forest fire research and fuel mapping in Europe.

The paper is very well written and full of suggestions and sharable concepts. The authors elaborate huge information and datasets but the results are not completely satisfactory. Unfortunately, at this stage the main innovation seems to be related with the integration of bioclimatic models to estimate the fuelbed depth in shrubland and grassland. However, because of the high variability of this kind of vegetation cover and the disturbances it is really difficult to validate the results of these models.

Response: We agree that one of the innovations of our manuscript is the usage of the bioclimatic models to estimate shrubland and grassland fuelbed depth in order to define fuel types. However, we would like to highlight other innovation aspects that may have not been seen because they were not explicitly mentioned in the manuscript or were not fully clarified.

- The paper proposes a new fuel classification system intended for European conditions and adapted to different fire management dimensions, such as risk, propagation, or emission estimations, among others. For this reason, the classification system includes two vertical strata (although only one has been mapped in this paper) and takes into account different canopy characteristic (leaf type and deciduousness). We consider that this is the first step towards a common and standardized fuel classification system in Europe that all countries and regions can use, which would help integrating fuel maps and would facilitate making comparisons among them. Other parts of the world, such as the United States, have a very long tradition in the use of fuel classification systems (the case of FBFM and NFFL).
- The generated European fuel map constitutes an updated fuel map for the European continent. Before this work was done, the only existing European fuel map was the EFFIS (European Forest Fire System, 2017) map from the year 2000 (22 years ago) that only mapped surface fuels using the NFFL system. Apart from this, the other fuel maps existing in Europe cover regional or local scales, or are land use maps rather than fuel model maps. So, the European fuel map generated in this paper updates the fuel information for the continent and serves for fire prevention and management purposes, mapping not only surface fuels but also canopy fuels. It is important to evidence that fuels are very dynamic in time, so having updated fuel maps is key to effective prevention activities (Keane et al., 2001; Keane and Reeves, 2012).
- The fuel classification system proposed in our manuscript is multiscale, meaning that the same system could be applied for fuel classification, not only at the European scale, but also at regional and local scales (one of the aims of the FirEUrisk project); therefore, it would help integrating fuel mapping in different areas and spatial scales. Having an overview of fuel distribution and characteristics in Europe is essential to have a holistic perspective that could help decision making and policies devoted to fire risk management. Moreover, the hierarchical nature of the classification allows to map at different levels of detail and at different scales depending on the available input data, but still they could be integrated and comparable because the fuel categories relate in the hierarchical scheme. The scalable system facilitates the harmonization of fuel maps in Europe.
- The proposed fuel classification system is intended to improve existing fuel classification systems by adding new categories that are key:
  - It considers the characteristics of both the forest canopy and understory. Some of the most used fuel classification systems (FBFM, NFFL) only refer to surface fuels. In the proposed classification, the authors tried to account for the different fuel strata so that the resulting maps

are useful for both surface and canopy fuels. Indeed, considering the characteristics of the forest canopy in the fuel classification (such as leaf type or fractional cover) could really help to identify areas where crown fires are most likely to occur.

- o It considers fuel categories referring to wet and peat/semi-peat land, which are key to fire emissions because of the amount of carbon and other greenhouse gases that they store, and that can be released to the atmosphere in case of a fire, thus contributing to climate change and affecting people's health (Weise and Wright, 2014; Van Der Werf et al., 2017; Zheng et al., 2021).
- o It considers urban fuels, defined in this paper as those fuels in the intermix or interface of built-up areas, referring to the Wildland Urban Interface (WUI). This constitutes an interesting improvement compared to pre-existing fuel classification systems where WUI areas are not considered fuels and therefore, are not parameterized (case of FBFM). Understanding urban fuels allows to assess residential and non-natural fuels, which can help to reduce fire risk to affecting human settlements and lives, and socio-economic losses (Bowman et al., 2017, 2020).

- The proposed fuel classification system is intended to be multipurpose. The lack of a common European fuel classification suitable for assessing different fire issues is a relevant matter. In this paper, expert knowledge among the project partners was used to identify the main needs of a fuel classification system to serve multiple purposes. That is why, the proposed fuel categories are defined to support fire prevention, assessment, propagation, behaviour, emissions, and suppression. It is important to highlight that, although in the paper the authors mapped fuel categories for the whole continent, the proposed fuel classification methodology can be replicated to map fuels at high-resolution, for local or regional scales (so that fire propagation models can be applied for local scales to model specific fires).
- The crosswalk to FBFM models constitutes a first approximation to fuel parameterization. As the reviewer knows, the regions or countries that have LiDAR data are relatively limited. Moreover, relevant issues in terms of resolution of LiDAR data and reference years of these data can be evidenced. Therefore, it is interesting to have a way to derive fuel parameters whenever there is a lack of existing resources. This is the idea behind the proposed crosswalk. The authors would like to highlight that the main results of section 4.2 are not the maps of parameters at European scale (this is an example of what can be done), but the crosswalk table that can be also applied for fuels that are mapped at higher resolutions.

References:

Bowman, D., Williamson, G., Yebra, M., Lizundia-Loiola, J., Pettinari, M. L., Shah, S., Bradstock, R., and Chuvieco, E.: Wildfires: Australia needs national monitoring agency, Nature, 584 (7820), 188–191, https://doi.org/10.1038/d41586-020-02306-4, 2020.

Bowman, D. M. J. S., Williamson, G. J., Abatzoglou, J. T., Kolden, C. A., Cochrane, M. A., and Smith, A. M. S.: Human exposure and sensitivity to globally extreme wildfire events, Nat. Ecol. Evol., 1, https://doi.org/10.1038/s41559-016-0058, 2017.

European Forest Fire Information System (EFFIS): European Fuel Map based on JRC Contract Number 384347 on the "Development of a European Fuel Map", European Commission [data set], 2017, https://effis.jrc.ec.europa.eu/applications/data-and-services, last access 21 May 2021.

Keane, R. E. and Reeves, M.: Use of Expert Knowledge to Develop Fuel Maps for Wildland Fire Management, in: Expert Knowledge and Its Application in Landscape Ecology, edited by: Perera, A., Ashton Drew, C., and Johnson, C., Springer Science+Business Media, New York Dordrecht Heidelberg London, 211–228, https://doi.org/10.1007/978-1-4614-1034-8_11, 2012.

Keane, R. E., Burgan, R., and van Wagtendonk, J.: Mapping wildland fuels for fire management across multiple scales: Integrating remote sensing, GIS, and biophysical modeling, Int. J. Wildl. Fire, 10, 301–319, https://doi.org/10.1071/WF01028, 2001.

Van Der Werf, G. R., Randerson, J. T., Giglio, L., Van Leeuwen, T. T., Chen, Y., Rogers, B. M., Mu, M., Van Marle, M. J. E., Morton, D. C., Collatz, G. J., Yokelson, R. J., and Kasibhatla, P. S.: Global fire emissions estimates during 1997-2016, Earth Syst. Sci. Data, 9 (2), 697–720, https://doi.org/10.5194/ESSD-9-697-2017, 2017.

Weise, D. R. and Wright, C. S.: Wildland fire emissions, carbon and climate: Characterizing wildland fuels, For. Ecol. Manag., 317, 26–40, https://doi.org/10.1016/J.FORECO.2013.02.037, 2014.

Zheng, B., Ciais, P., Chevallier, F., Chuvieco, E., Chen, Y., and Yang, H.: Increasing forest fire emissions

despite the decline in global burned area, Sci. Adv., 7 (39), https://doi.org/10.1126/SCIADV.ABH2646/SUPPL_FILE/SCIADV.ABH2646_SM.PDF, 2021.

We have summarized this information in the second paragraph of the conclusions: "The results of this study constitute the first step towards a risk-wise landscape and fuel mapping development across Europe, which will help integrated, strategic, coherent, and comprehensive decision making for fire risk prevention, assessment, and evaluation. The results have wide applicability because they meet the actual unfulfilled fuel mapping needs in Europe: 1) the development of a fuel classification system specifically designed for European conditions, which allows not to rely on external classifications that should be only applied to the regions for which they were developed, 2) enabling coordination, integrating fuel mapping at different spatial scales and across European regions through a common fuel legend with hierarchical levels, 3) multipurpose, including prevention, propagation, behaviour, emissions, and suppression, 4) mapping fuel types not previously considered at European scale that are key for protecting people and the environment from the devastating effects of fires: forest canopy fuels (key for crown and extreme fires), wet and peat/semi-peat land fuels (key for emissions) and urban fuels in the Wildland Urban Interface (key for people's and socio-economic safety), 5) the generation of an updated European-specific fuel map, compared to the EFFIS fuel map from year 2000 (European Forest Fire Information System (EFFIS), 2017), and 6) the preliminary surface fuel parameterization for Europe that can be used for estimating fuel parameters whenever there is no suitable input data available. Overall, the existence of updated land cover datasets and bioclimatic models for the European territory is limiting, and work is still needed to parameterize canopy fuels. The results of this work are part of the new FirEUrisk integrated three-part perspective of fire risk, whose strategy is meant to shift the thinking of wildfire management by looking simultaneously to fire assessment, reduction, and adaptation from a common scheme."

Starting from the classification, the considered forest classes are very similar to existing vegetation cover maps already available. Using a threshold of 15% of trees does not allow to discriminate fire behaviour if there is no information on which kind of fuel is present in the 85% of the area. In addition, it is evident that some species are more flammable than others and the considered classes does not allow to discriminate among these species.

Response: The reviewer is right, fuel classification systems have an important relation with vegetation cover classifications because fuel types are vegetation types, which are mapped in many land and vegetation cover maps. The difference between fuel types and land and vegetation cover types is the use of a fire perspective to map and assess the first ones. Therefore, although they are similar concepts, they are also different. An example of this is that in fuel maps there is no interest in mapping non burnable categories, while land and vegetation cover maps map all types of covers. Likewise, fuel classification systems define their categories by distinguishing vegetation types that present similar characteristic fire behaviour or cause similar effects (such as emissions), while many times land and vegetation cover classifications discriminate categories by characteristics that are not related to fire spread and behaviour such as socio-economic, historical, or conservation aspects (e.g. protected areas, exploited forests, historical forests, different types of agricultural uses, etc). In order to better prevent fire risk in Europe, it is preferable to have maps adapted to this purpose (fuel maps) instead of general maps not specifically adapted to fire issues (land and vegetation cover maps).

Regarding the second comment about the threshold to identify forests, the authors agree with the reviewer on the fact that the used thresholds are key to how the final maps would look like and determine which categories have more importance in the classification. In any case, the selection of a threshold for determining categories is based on the best compromise among different information sources, classifications, situations, and inherent limitations. Moreover, as the reviewer knows, the selection of a threshold other than 15% (i.e.: 50%; 30%; or what else) would have resulted in some other strengths or weaknesses: in other words, there is not a threshold value that can be adopted, or even suggested by an expert, without having some potential critical aspects and limitations. In our case, the use of 15 % as threshold to define forest fuels was based on expert knowledge and relates to a common definition of forest, as the one of the Copernicus Global Land Cover legend (Tsendbazar et al., 2020) based on the UN-LCCS (United Nations Land Cover Classification System) from the UNESCO (United Nations Educational, Scientific and Cultural Organization) (UNESCO, 1973) and the FAO (Food and Agriculture Organization, 2000). Information on this has been added in new line 140: "Forest: areas with tree canopy cover above 15 % with a mean tree height ≥ 2 m, following the Copernicus Global Land Cover legend (Tsendbazar et al., 2020), which is based on the UN-LCCS (United Nations Land Cover Classification

System) from the UNESCO (United Nations Educational, Scientific and Cultural Organization) (UNESCO, 1973) and the FAO (Food and Agriculture Organization, 2000)". Moreover, this "low" threshold is conservative to account for the presence of forest fuels, which helps to identify possible spots of crown fires occurring, the most severe. However, it should be considered that any fuel classification system has the limitation of generalizing the main fuel characteristics, and mapping fuel types has the limitation of mixed covers. This is a common issue for categorical maps (what fuel maps are), as it was already identified by Pettinari and Chuvieco (2016) in their global fuel map, but it also affects biomass and cover maps at continental and global scales.

Regarding the lack of information about the remaining area not covered by forest, the second-level of our proposed fuel classification system would enable to obtain this information should the available data allow it.

Although flammability is key to fire ignition, it is not generally considered in fuel models, which mainly consider geometrical properties of the vegetation (particle size, vertical-horizontal distribution, etcetera). Also, fuel models do not generally discriminate vegetation species, but rather those vegetation groups that have similar combustion properties. One of the purposes of fuel maps is to map fuel types according to these characteristics, indeed. This is done by grouping fuel species potentially having similar behaviour during a fire (e.g., the leaf type, deciduousness, and fractional cover forest groups, or the groups by fuelbed depth). In fact, our classification system distinguishes more fuel types than the most commonly used standard fuel classification systems such as the NFFL (Anderson, 1982), FBFM (Scott and Burgan, 2005), or Prometheus (European Commission, 1999; Arroyo et al., 2008). In addition, it is very complicated to map species at the European scale because of the high heterogeneity of forests and the actual capabilities of multispectral sensors. Of course, the authors agree on the fact that this kind of information would enrich the derived maps and models.

As a general consideration, I suggest to remove the paragraph 4.2 and define the association of fuel models parameters in a second stage of this work when most of the presented issues will be solved using other technologies such as Lidar, AI, etc…

Response: Section 4.2 refers to fuel parameterization, which is currently on-going research. We are working on developing custom maps of fuel parameters at the European scale through calibrating models integrating LiDAR, multispectral and SAR data using a Machine Learning framework. However, we consider useful for the reader to show how a standard fuel model (in this case, the FBFM models) could fit to our FirEUrisk fuel types. This way, the proposed crosswalk ensures having fuel parameters for all areas in Europe. The authors would like to highlight that the main achievement of section 4.2 are not the maps of parameters at

the European scale (this is an example of what can be done), but the crosswalk table that could serve to facilitate the use of our fuel classification system whenever LiDAR or field data and ML or AI models are not available. As said, we are working on improving fuel parameterization in a second stage, so we believe that it is interesting to present in this paper a first assignment of fuel parameters.

The paper can be accepted after major revision and a better description of the objective of this work starting from the title, the abstract and the conclusions. This could help also to make the paper shorter, improving its readability.

Response: Thank you for the suggestions. We agree on the need of further clarification of the objectives and innovation of our work, so changes have been done accordingly. The novelty of the work is now better emphasized, and the paper has improved its readability. For the abstract, the three working-steps where clarified: 1) the proposal of the new adapted-to-Europe fuel classification system, 2) the generation of a European fuel map using the proposed fuel classification system, and 3) the fuel parameterization. It has also been clarified the difference between the whole proposed fuel classification system and the fuels that were mapped at the European scale (only the first-level of the fuel classification system because of the working scale and available data). This last issue has been clarified throughout the text with sentences such as (new line 181) "In this paper, the European fuel map was only based on the first-level of the proposed fuel classification system". For the conclusions, the key innovation aspects of the work have been added, which are the summary of the ones reported in the second answer of this document. Also, to clarify the objective, Figure 1 has been improved as follows:

[Figure]

**Figure 1.** General overview of the structure of this work.

[revised manuscript text omitted]

The authors thank the reviewer for his/her comment on the title of the manuscript, but we would like to keep the title as it is.

**Responses to the specific comments:**

Line 21: It is not clear the difference between the conceptual development and the resulting fuel classification system. Reading the paper the ambitions presented in the abstract seams to be not satisfied. Please, rewrite the abstract according to the obtained results.

Response: The concepts have been clarified by changing some of the abstract sentences (see our answer above).

Line 64: Depending by the vegetation type. I'll suggest the authors to concentrate their classification to identify which kind of forest have an higher probability of transition from ground to crown fire. This could be a great improvement to the current land cover map available.

Response: This is a good point, as assessing the potential transition from surface to crown fires is key to prevent crown fires. In the same line, the authors believe that it is important to both identify those forests that can easily go from surface to canopy and those that cannot. We agree with the reviewer, and in fact, forest canopy and understory categories are proposed in the fuel classification system, Also, as the reviewer knows, crown fires are highly influenced by the characteristics of understory and ladder fuels, as well as by wildfire intensity (e.g., flame length), information that is not available at the European scale. However, we encourage to complement the proposed fuel types with additional data for the regions where it may be available. In other words, evaluating the likelihood that crown fires can occur would require determining the vertical continuity of forests, as well as identifying the existence (or not) of a gap between the understory and the canopy fuels, and estimating potential fire behavior, which is the result of complex relations among weather, topography, and fuels. Regarding the last point, for instance, fire behavior can be estimated using wildfire spread models and considering the conditions associated to fire occurrence in a given area. Overall, the point highlighted by the reviewer is intended to be analyzed by the authors in future works.

To clarify this, we have added to the text the following in new line 587: "The predicted increase in fire intensity and occurrence of the so-called megafires (San-Miguel-Ayanz et al., 2013), which usually evolve from surface to crown fires, makes it necessary to improve our information on canopy fuels. Assessing the potential transition from surface to crown fires is key to prevent crown fires. For this reason, our classification approach includes both surface and canopy fuel types for the forest fuel types. Crown fires are highly influenced by the characteristics of understory and ladder fuels, as well as by wildfire intensity (e.g.: flame length), information that is not available at the European scale. However, we encourage to complement the proposed fuel types with additional data for the regions where it may be available. This would require determining the vertical continuity of fuels, as well as identifying the existence (or not) of a gap between the understory and the canopy fuels strata. This might be subject of in future work. The rest of the fuel types are disaggregated based on their fuelbed depth, with thresholds suggested by the experts. However, fuel mapping is still a challenge because of the high spatiotemporal variability of fuels, and the need to generalise the great variety of vegetation conditions related to fire behaviour."

Line 80: I completely agree with this sentence but it is not clear how the classification porposed overcome this limitiation.

Response: The proposed fuel classification system is a first attempt towards filling this gap (fuel classification systems only referring to surface fuels), which the authors plan to keep improving in future work. For the moment, what has been achieved is the proposal of a fuel classification system that differentiates among dissimilar forest canopies and understory types. This is a first attempt to try to differentiate surface from crown fires as both characteristics have been considered separately (and can be mapped), and which could also help study the potential transferability of fire from surface to crowns. We agree with the reviewer that this is still a challenging issue that needs improvement to prevent crown fires in future works.

Line 93: Global scale include also Europe. Please, explain the limitation and the innovation for the European-level.

Response: The reviewer is right as the global scale includes Europe. However, applying a global map to the European scale has its limitations, for which the authors tried to account in the present paper.

We modified the sentence as follows in new line 89: "Fuel maps exist for continental scales, such as South America (Pettinari et al., 2014) and Africa (Pettinari and Chuvieco, 2015); and global scales, but including categories that are too coarse to be operationally applicable to European conditions (Pettinari and Chuvieco, 2016)."

This information was completed in the discussion section as follows in new line 564: "Anyhow, the obtained results constitute an improvement in European fuel mapping compared to existing fuel maps covering the European territory. The map provides more detailed categories than those of existing global fuel maps (Pettinari and Chuvieco (2016)), or the 2000 EFFIS fuel map (European Forest Fire Information System

(EFFIS), 2017), which only referred to surface fuels, thus not considering forest canopy characteristics. In addition, the FirEurisk fuel map includes new categories such as wet and peat/semi-peat land fuel types, which are key to understand fire emissions; and urban fuel types, crucial to prevent fire affecting humans, which were not considered in previous continental and global fuel maps."

Line 100: Please, try to better explain how the proposed classification fill this gap.

Response: This aspect was explained modifying the discussion section as follows starting in new line 559: "The proposed FirEUrisk hierarchical fuel classification system was designed to be adapted to a wide range of environmental conditions, including those found in the European territory, describing both surface and canopy fuels. In this paper, we present a first product based on this classification, covering the whole European territory for the first-level of the classification. We did not consider the forest understory, second-level of the classification, better suited to regional and local scales where more detailed information, particularly LiDAR data, can be available. Anyhow, the obtained results constitute an improvement in European fuel mapping compared to existing fuel maps covering the European territory. The map provides more detailed categories than those of existing global fuel maps (Pettinari and Chuvieco (2016)), or the 2000 EFFIS fuel map (European Forest Fire Information System (EFFIS), 2017), which only referred to surface fuels, thus not considering forest canopy characteristics. In addition, the FirEurisk fuel map includes new categories such as wet and peat/semi-peat land fuel types, which are key to understand fire emissions; and urban fuel types, crucial to prevent fire affecting humans, which were not considered in previous continental and global fuel maps.
The hierarchical nature of the system aims to define a common fuel types' classification for different scales and study areas. It also offers high versatility, as it enables mapping fuels with different disaggregation of categories, depending on the detail and quality of the input data, while allowing to overlap fuel maps for the same area at different scales, which would help the integration and comparison of fuel maps because of the common legend. Thus, whereas the fuel map developed at the European scale was based on existing European and global datasets integrated into a GIS framework, the same classification scheme could be applied to provide a more comprehensive fuel classification using a multi-sensor approach in a machine learning framework (García et al., 2011; Marino et al., 2016; Domingo et al., 2020). Its structure has similarities (e.g., hierarchical scheme) with the ArcFuel classification (Toukiloglou et al., 2013), although this was only prepared for southern-European conditions. In addition, the involvement of expert knowledge in the development of the FirEUrisk hierarchical fuel classification system suggests high acceptance, and therefore usage, among the fire risk management community in the foreseeable future. It also allowed the development of a useful classification, intended to fill the actual gaps of the European fuel mapping, towards a homogeneous and integrated fire risk prevention strategy. Nevertheless, it must be considered that the grouping of vegetation types into fuel types is a balance between generalisation of the landscape reality and loss of detailed information, which may not be the most suitable system for all study areas."

Also, the last paragraph of the introductory section was modified adding some sentences to clarify how identified gaps are filled: "Considering the current limitations of European fuel mapping, this paper had three objectives. The first one was generating a fuel classification system to facilitate the integration of continental wildfire risk assessment, including both surface and canopy fuel types. The proposed classification system should be hierarchical to facilitate the integration of fuel maps at different spatial scales, include both surface and canopy fuel types and be suitable for different purposes, from fire behaviour simulation to fire emissions or fire danger assessment. The second objective was to develop a European fuel map at 1 km spatial resolution following the proposed fuel classification system. We aimed to develop a methodology that, combining expert knowledge, GIS, available datasets, and bioclimatic modelling, might be easily replicable and updated with low time and economic costs. Finally, the third objective was to assign surface fuel parameters to the derived fuel types, by relating them to existing fuel models. We chose the FBFM standard fuel models (Scott and Burgan, 2005), as this system is widely used and very flexible. These three objectives serve to organise the structure of this paper around three sections (Fig. 1). This work is expected to lay the framework for an integrated and homogeneous fire management strategy across European countries. The present study is part of the FirEUrisk project, which aims to create a European integrated strategy for fire danger assessment, reduction, and adaptation."

Line 102: was.

Done.

Line 111: Please, better justify the choice of 1 km resolution. It is evident that this resolution is not suitable for the described purposes.
Line 342: It is not clear why the map at 100 m resolution has been reprojected to 1 km. This resolution strongly limit the use of the map for any fire management purposes. Please, explain why you decide that the target spatial resolution is 1 km.

Response to both lines 111 and 342: Although this 1 km resolution might be seen as coarse, it should be noted that the study area is the European continent. The extent of this study area and the available input data makes it difficult to map fuels at a finer scale, especially considering the input data (the CCI LC map at 300 m, the bioclimatic models (including the weather data), etcetera), which led to adopt 1 km as a compromise resolution. It should be considered that the intention to generate a European-scale fuel map is not to simulate fire propagation at regional or local scales, but to have an overview and a holistic perspective of the actual state of fuels in the European territory, that would help to identify areas with higher fire risk conditions and where specific prevention strategies might be adopted to reduce fire risk. In addition, the proposed fuel classification system is hierarchical and multipurpose, referring not only to propagation studies but also to emissions or post-fire recovery, for which 1 km resolution could be a first step to compare fuel and fire risk conditions at a continental scale. Anyway, the classification can be adapted to different spatial scales and in fact, the same legend is being used for Pilot Sites in Europe (regional scales at the FirEUrisk project where the target resolution is 1 Ha or higher), so it has a scalable structure. Nevertheless, similar methods could be developed to map fuel maps at higher resolution for the European scale in future works.
We have added the following to the manuscript in new line 349: "The input layers used for the generation of the European fuel map were previously resampled to 100 m to match the spatial resolution of the Copernicus GLC map, which was our main information source. However, the spatial characteristics of some of the input layers (such as the CCI LC map at 300 m, and the bioclimatic models based on 1 km resolution weather data), recommended to convert the final product to 1 km spatial resolution, which was also the project target resolution for the European scale. Therefore, after obtaining the first fuel type dataset at 100 m resolution, it was resampled to 1 km, carefully accounting for the heterogeneity of European fuel types."

Line 132: Educational,

Done.

Line 133: ,

Done.

Line 139: Why urban areas are considered as fuels? Does it is part of the innovation in the classification proposed? If yes, please explain how you characterize fuel parameters for this fuel class.

Response: Yes, some anthropic areas are considered fuels, and this is part of the innovation in the proposed fuel classification system, as it was explained in the second answer to the reviewer in this document. The following information was added to explain this in new line 161: "Urban: areas with ≥ 15 % built-up structures and/or buildings. The standard CLC division between continuous and discontinuous fabric was followed, related to the amount of vegetation belonging to the intermix and interface of the Wildland-Urban Interface (WUI). This is part of the innovation of the proposed classification system, as it allows the assessment of residential and non-natural fuels, which can in turn help identifying anthropic areas where fires can affect human settlements and lives."
Fuel parameters for this fuel category are pretended to be extracted in the same way as the rest of fuel categories because they are also vegetation, just being surrounded by buildings and infrastructures.

Line 141: Please, better explain how the two vertical strata were defined.

Response: To do this, the paragraph was modified as follows in new line 169: "Forest categories were divided into two vertical strata: the first-level referred to the overstory (canopy) characteristics, and the second-level to the understory characteristics. Further subdivisions were included in the first-level by considering the leaf type (broadleaf/needleleaf), the leaf deciduousness (evergreen/deciduous), and the fractional cover (open/closed). The lower stratum referred to the understory characteristics by identifying the type of surface vegetation (grassland/shrubland/timber litter), and its height. This allowed us to define the surface and canopy characteristics of the fuels in the forest, which can help to account for both surface and crown fires."

Line 142: phenology has a different meaning.

"Phenology" has been changed by "Leaf deciduousness (evergreen/deciduous)" throughout the text.

Line 149: Does it mean that these classes are defined only for high spatial resolution? Please, clarify.

Response: Yes, the reviewer is right. For the European scale, it is only intended to map a general nonfuel category, as it was considered that there was no need to discriminate among different types of nonfuel land covers (it does not report any information usable to assess fire behaviour). For higher scales, nonfuel cover types might be discriminated if it is interesting for any reason. This has been clarified as follows in new line 166: "Nonfuel: permanent water bodies, open sea, snow, ice, bare soil, sparse vegetation (< 10 %). It was not found relevant to further disaggregate non-fuels by mapping water, snow, ice, bare soil, and sparse vegetation, but it could be easily introduced if desired at high spatial resolutions."

Line 156: How the threshold of 15 % was selected? It is extremely difficult to discriminate fire behaviuor based on this threshold and it is one of the main criticality of the already existing fuel map. How this new classification overcome this limitation?

Response: This issue was deeply explained in the third answer to the reviewer in this document. To summarize it, the following was added to the paper in new line 140: "Forest: areas with tree canopy cover above 15 % with a mean tree height ≥ 2 m, following the Copernicus Global Land Cover legend (Tsendbazar et al., 2020), which is based on the UN-LCCS (United Nations Land Cover Classification System) from the UNESCO (United Nations Educational, Scientific and Cultural Organization) (UNESCO, 1973) and the FAO (Food and Agriculture Organization, 2000). "

Line 171: Please, see previous comments on this.

Response: See previous answer on this.

Line 177: Please, explain how the non fuel categories are discriminated for the second level.

Response: The sentence referring to this was deleted because it caused confusion. Second-level fuels are not mapped at the European scale, so no discrimination was done. This was clarified in new line 181: "In this paper, the European fuel map was generated for the first-level of the proposed fuel classification system, covering all European continental countries at 1 km spatial resolution."

Line 184: Could you explain why this new map would help the strategic planning of fire management in Europe?

Response: Thank you for the suggestion. This was explained by adding the following in new line 182: "This product was developed to help the strategic planning of fire management in Europe through generating a continental map with a homogeneous and integrated fuel classification system for all countries, which would allow to carry out standardized fire risk analysis and inform fire managers and policy makers from a risk-wise holistic perspective for Europe."

Table 1: The class mixed forest has been always the most critical in terms of fire behaviour prediction because it is difficult to understand if part of the vegetation is represented by flammable species or not. How this classification help to overcome this limitation?

Response: The authors agree with the reviewer on the idea that having detailed information on forest types would help fire behaviour prediction. However, there would be forests where the mixture of species and tree types would be that high that they should be considered as mixed. Mapping categories at non-very-detailed resolution would always require a sort of generalization of the reality. This has been identified as a limitation of categorical maps (as explained in previous comments). Anyway, the proposed classification tried to overcome this limitation by defining fuel types with similar characteristics (leaf type and deciduousness, fractional cover, and understory type and depth) that are expected to behave similar in case of fire. In fact, this level of disaggregation of forest fuels is higher than those from existing fuel classification systems (FBFM, NFFL, Prometheus, etc). This way, although not fully, the authors tried to develop a fuel classification system that grouped fuel types in the most disaggregated categories as possible. Only in the case of very mixed forest where it would not be true and accurate to identify them as a certain forest fuel type, it was used the category of mixed forest.

Line 197: winter season is an important fire season in several areas. Fires in spring and autumn are related with the extension of summer and winter season due to climate change and interannual variability. Please, rewrite this sentence.

Response: We rephrased the sentence in new line 192 according to the reviewer's suggestion: "The study area is the European territory as defined by the FirEUrisk project, with around 5 Mkm$^2$ of land, covering 33 countries (Fig. 2). The most historically affected European countries by wildland fires have been Portugal, Spain, Italy, Greece, and France. However, a recent increase in fire activity in higher latitudes has been observed: e.g., fires in Sweden in 2018 (San-Miguel-Ayanz et al., 2021), and the fire between the Czech Republic and Germany in 2022 (Global Disaster Alert and Coordination system, 2022). The most dangerous fire conditions in the European territory, and particularly in the most affected Southern European Union countries, are usually observed during the summer months, which represent the period where fuel conditions are most favourable to fire ignition and spread. The peak of the fire season can be different in other European areas, observed in winter (e.g., Alps; Pyrenees) or spring (Central and Northern Europe) (San-Miguel-Ayanz et al., 2021)."

Line 200: It is not clear why burned areas in the considered period are included in this map. I suggest to remove burnt area from figure 2. Maybe you could include an additional picture of burned areas in winter season compared with summer season.

Response: The suggestion of the reviewer was accepted. The original map was deleted and a new map was included:

[Figure]

**Figure 2.** Study area, and burnt areas from 1 January 2000 up to 27 January 2022 in winter and summer seasons (EFFIS, 2021).

Line 259: How?

Response: Leaf type and deciduousness of the unknown forests from the Copernicus GLC map, was determined by intersecting the Copernicus GLC map with the forest categories of the CCI LC map. This is explained in the following lines (paragraph starting in new line 258): "Information on the leaf type, leaf deciduousness, and fractional cover of forest fuels was obtained from the Copernicus GLC map (Buchhorn et al., 2020). This dataset defines all the first-level forest fuel types in the FirEUrisk fuel classification system, plus two more categories only referring to fractional cover: unknown open forest and unknown closed forest. Pixels falling in these two categories were overlapped with the CCI LC map (Copernicus Climate Change Services, 2020), previously resampled from 300 m to 100 m using the nearest neighbour method to match the resolution of the Copernicus GLC map. This allowed determining the leaf type (broadleaf/needleleaf) and leaf deciduousness (evergreen/deciduous) of the unknown forest from the CCI LC map for forest cover. The pixels identified as unknown forest in the Copernicus GLC map but not as forest in the CCI LC map were assigned the category of the CCI LC map."

Line 270: The use of bioclimatic models for estimating grassland and shrubland fuelbed depth seams to be one of the main innovation of the proposed classification. However, it is not clear how the models have been selected and how the results have been validated. Please, try to better explain this aspectes. Have the selected models been adapted to Europe?

Response: The usage of the bioclimatic models to estimate fuelbed depth is indeed a very innovative aspect of this work. To the authors' knowledge, there do not exist bioclimatic models to extract fuelbed depth for the whole European territory, but instead there exist some papers regarding this issue for local scales. For the generation of the map, we decided to apply the already-existing bioclimatic models best matching European conditions. In the case of shrubland, because most of the shrubland fuels in Europe were located in the arid/semi-arid areas (Table 4), bioclimatic models for shrublands in these areas were used. These models were not adapted to Europe (except the data that gets the productivity of the grasslands because it was developed for the European scale) because of the lack of calibrating methods for such a big area. We are aware that this approximation has its limitations, but the available input data for such a big area as Europe and the calibrating data available did not allow to calibrate own bioclimatic models. These models are already validated, and their details can be read in the papers where they were originally presented. We have provided detailed information in the new version of the manuscript in new line 279: "There do not exist bioclimatic models adapted to the whole European conditions, so we used the regional already-calibrated models which best related to European shrubland conditions (mostly located in arid-semi arid zones) as an approximation."

Line 350: how they are discriminated?

Response: The paragraph (new line 359) was rewritten to clarify how dominant categories in a 1 km pixel were discriminated: "The main resampling criterion was to choose the dominant (first-mode) category within the target pixel. However, to tackle the impact of mixed fuel type covers (e.g., mixed forest), and to take into account the most dangerous type between two equally-extended fuel types (discriminated using expert knowledge); the combination of categories in Table 2 was performed whenever there were two co-dominant categories. Co-dominant categories were defined as those that present the same frequency in a group of 10 x 10 pixels, or the frequency of one category is higher than half the frequency of the other category. The combination of the co-dominant categories in Table 2 was carried out regardless of which category had higher frequency."

Line 375: did you make it over 5,016 points?
Line 381: why you did ot make use of all the points selected? Please, justify.

Response to both lines 375 and 381: The validation procedure was clarified as filtering was not performed to 5,016. Filtering was made to all LUCAS points and finally, after filtering, the validation was done using 5,016 points, which were considered to be a sufficiently representative sample according to the proportion of area covered by each fuel category. This has been clarified in the text by deleting the confusing data and modifying the following sentences (new line 394): "Finally, after applying the filters we extracted 5,016 suitable LUCAS validation points by stratified random sampling, which was considered a representative sampling according to the proportion of area covered by each fuel category. The land cover categories from the validation points were reclassified to the most similar FirEUrisk main fuel types and were used for the assessment of the European fuel map. A confusion matrix was computed for quantitative analysis."

Line 401: Where are the results of this comparison? I was not able to find any comparison in the paper. Please add a table or a plot to show this comparison.

Response: The comparison between the validation of the FirEUrisk European fuel map and the confusion matrices of the 2015 Copernicus GLC map over Europe is performed in the discussion section, as this work results are being compared to other existing datasets and results. To make it easier to find this comparison, it has been specified in the methods section in new line 414: "Finally, in the discussion section, the two confusion matrices (one for the main fuel types, another for all mapped fuel types) were compared to the results obtained from the validation of the 2015 Copernicus GLC map over Europe (Tsendbazar et al., 2020)." The comparison is made through the text because generating a table or plot would be difficult to understand, as the categories in both datasets are not the same and only some of them have an "equivalent".

Line 420: Are you sure that the model for shrubland fuelbed depth is adapted to Europe? Is it possible to consider shrubland up to 6 meters?

Response: Regarding the first question, the shrubland model has two steps. In the first step, the model to estimate shrubland biomass from Mean Annual Precipitation data developed for the State of California was used. In the second step, the model to estimate shrubland fuelbed depth from shrubland biomass developed for Turkey was applied. Because European shrublands were identified to predominate in the arid/semi-arid areas of the European continent (mostly represented by the Mediterranean biogeographic region - European Environment Agency (2016)), it was found appropriate to use the models developed to California and Turkey, as their climatic conditions are like those in the arid/semi-arid Europe. This was seen in the ClimateCharts.net platform (Zepner et al., 2020).
Regarding the second question, the following was added in new line 430 and is self-explicative: "Although shrubland are generally considered up to 5 m, exceptions are allowed subject to the plant's physiognomic aspect (Food and Agriculture Organization, 2000). Therefore, here we allowed for plants higher to 5 m being classified as shrubland if they have a clear physiognomic aspect of shrub."

Line 552: Please, specify that no information about the vertical distribution of vegetation is included in the forest classification.

Response: Done through this sentence in new line 561: "In this paper, we present a first product based on this classification, covering the whole European territory for the first-level of the classification. We did not consider the forest understory, second-level of the classification, better suited to regional and local scales where more detailed information, particularly LiDAR data, can be available."

Line 553: Try to better define the improvement with respect to the global fuel map.

Response: Done through the following sentences starting in new line 564: "Anyhow, the obtained results constitute an improvement in European fuel mapping compared to existing fuel maps covering the European territory. The map provides more detailed categories than those of existing global fuel maps (Pettinari and Chuvieco (2016)), or the 2000 EFFIS fuel map (European Forest Fire Information System (EFFIS), 2017), which only referred to surface fuels, thus not considering forest canopy characteristics. In addition, the FirEurisk fuel map includes new categories such as wet and peat/semi-peat land fuel types, which are key to understand fire emissions; and urban fuel types, crucial to prevent fire affecting humans, which were not considered in previous continental and global fuel maps."

Line 558: it is not clear how the classification can be scaled and how the categories can be disaggregated.

Response: This has been clarified by adding sentences to the referred paragraph starting in new line 571: "The hierarchical nature of the system aims to define a common fuel types' classification for different scales and study areas. It also offers high versatility, as it enables mapping fuels with different disaggregation of categories, depending on the detail and quality of the input data, while allowing to overlap fuel maps for the same area at different scales, which would help the integration and comparison of fuel maps because of the common legend."

Line 603: Again, it is not clear the advantage to consider urban fuel type.

Response: This aspect has been already improved in previous comments and parts of the paper. Thus, this paragraph was complemented with the same information as before in new line 626: "Urban fuel types are the least represented in Europe, but they are the most dangerous from an economic, societal and human health point of view (Bowman et al., 2011). Mapping urban fuel types represents an advance of the proposed classification system, as it allows the assessment of residential and non-natural fuels, which can in turn help identifying anthropic areas where fires can affect human settlements and lives."

Line 616: This is the only information on the comparison with 2015 Copernicus GLC map. It is not clear how the proposed approach would improve the usability of the map.

Response: In this part or the paper, we compared the validation report of the Copernicus GLC map and this paper's validation results. The authors did not intend that this improves the usability of the map.

Line 688: Please, Try to list some of the insights.

Response: Done by adding them in new line 711: "The FirEUrisk fuel classification system can provide a number of insights and information for wildfire risk monitoring and assessment at the European scale including fuel parameters, such as dead and live surface fuel load, Surface to Area Volume ratio, or surface fuelbed depth."

Line 691: Most of the information needed for running propagation models is not present in this work.

Response: Thank you for pointing out this limitation. What we obtained was the fuel information needed to run fire propagation models, but then, we need to input other data (e.g., weather conditions, topography,

ignitions, etc) to make the run. We have corrected the sentences in the text to reflect this in new line 716: "The full surface fuel set information needed to run fire propagation models can be extracted from the crosswalk to the FBFM, complemented with other canopy fuel parameters (such as crown base height or crown bulk density) and other necessary input data (e.g., weather conditions, topography, ignitions, etcetera) to run fire spread models (e.g., FlamMap (Finney, 2006) and FARSITE (Finney, 2004), as embedded in FlamMap 6.2 (https://www.firelab.org/project/flammap). This should be subject of an extension of this paper and could be based on the calibration of models that estimate canopy fuel parameters using airborne and satellite LiDAR systems, for which regional airborne LiDAR would be key to consider the heterogeneity of European fuels before using the global satellite LiDAR data for the continental scale."

Line 699: Please, try to better explain how this map help to rate fire danger and risk conditions.

Response: Done by adding in new line 727: "In addition, the FirEurisk fuel map would be useful for regions that do not have fuel cartography. The mapped fuel types and the fuel parameters obtained from the crosswalk to FBFM can serve as input for fire propagation models and help rate fire danger and risk conditions. It is also important to note that the maps of fuel parameters at the European scale are examples of what can be done, but the crosswalk is intended to be useful for areas where technologies and resources such as LiDAR data are not available."

Line 728: It is really unclear how the results of this work meet the unfulfilled fuel mapping needs in Europe. Conclusions need to be expanded and better define the results of this work.

Response: We agree with the reviewer that the original version of the paper was unclear in this aspect. We have deeply modified the last paragraph of the conclusions to summarise how this work tries to meet the actual unfulfilled fuel mapping needs in Europe: "The results of this study constitute the first step towards a risk-wise landscape and fuel mapping development across Europe, which will help integrated, strategic, coherent, and comprehensive decision making for fire risk prevention, assessment, and evaluation. The results have wide applicability because they meet the actual unfulfilled fuel mapping needs in Europe: 1) the development of a fuel classification system specifically designed for European conditions, which allows not to rely on external classifications that should be only applied to the regions for which they were developed, 2) enabling coordination, integrating fuel mapping at different spatial scales and across European regions through a common fuel legend with hierarchical levels, 3) multipurpose, including prevention, propagation, behaviour, emissions, and suppression, 4) mapping fuel types not previously considered at European scale that are key for protecting people and the environment from the devastating effects of fires: forest canopy fuels (key for crown and extreme fires), wet and peat/semi-peat land fuels (key for emissions) and urban fuels in the Wildland Urban Interface (key for people's and socio-economic safety), 5) the generation of an updated European-specific fuel map, compared to the EFFIS fuel map from year 2000 (European Forest Fire Information System (EFFIS), 2017), and 6) the preliminary surface fuel parameterization for Europe that can be used for estimating fuel parameters whenever there is no suitable input data available. Overall, the existence of updated land cover datasets and bioclimatic models for the European territory is limiting, and work is still needed to parameterize canopy fuels. The results of this work are part of the new FirEUrisk integrated three-part perspective of fire risk, whose strategy is meant to shift the thinking of wildfire management by looking simultaneously to fire assessment, reduction, and adaptation from a common scheme."

Please, find also attached the new version of the manuscript with the changes done.

---

## Author Comment (AC3)

**Response to Anonymous Reviewer #3**

Dear authors,

I was carefully reading the manuscript several times and I found the manuscript well structured and written, with very interesting findings. Indeed, you used very updated and deep literature on the topic and clearly, you added new inputs adapting these fuel maps approach in a European context, being the main gap. However, there are some points that I would like to stress, which could improve the quality of the manuscript. See the comments below:

Response: The authors thank the reviewer for taking the time to review our manuscript. We also appreciate very much the advises and suggestions made. We have thoroughly considered the suggestions made. Improvements have been made in order to improve the quality of the manuscript.

**General comments**

Introduction

The introduction demonstrated a deep state of the art regarding fuel classification systems, globally and continentally. The review about fuel classification is a great input for other researchers, where the authors put a lot of effort, being a very valuable contribution for future investigations. Well done!

Even though the authors demonstrate a great knowledge and handling of the literature on the subject. The introduction is extremely long and repetitive. From what I understood, subchapters 2.1 and 2.2 are also part of the introduction, or? I would suggest sending these two subchapters as supplementary material, since they are part of a technical report rather than part of a scientific article.

Response: We thank the reviewer for his/her positive comments on the introduction. We are aware that the structure of the manuscript might be a bit complex because the paper has several parts. Introduction is only section 1, not including subchapters 2.1 and 2.2, which refers to the first out of three objectives of the paper: the conceptual development of the new proposed fuel classification system. To clarify and simplify the structure of the whole manuscript, we have unified subchapters 2.1 and 2.2 as "2 Design of the FirEUrisk hierarchical fuel classification system". Moreover, to take into account the reviewer concerns, we shortened the text and deleted some redundant sentences from the Introduction and section 2.

Also, the objectives section in the introduction has been clarified by detailing the workflow in Figure 1 as follows, highlighting the three main parts of the paper:

[Figure]

**Figure 1.** General overview of the structure of this work.

Material and methods

The methodology in general is well addressed. However, I have the following concerns:

- The European fuel classification map that you created to which year belong 2016, 2018, 2019 or 2020? Because it is important to define to which year the map belong independently of the year of publication (article). I would suggest use the oldest layer input to define the year of the European fuel map classification.

Response: The used input land cover datasets for the generation of the European fuel map are the Copernicus GLC map (2019), CCI LC map (2020), CLC map (2018) and Built-up fraction cover Copernicus GLC map (2019). The main used input layer is the Copernicus GLC map (2019), that is why we stablished that our product is circa 2019. This was also added in the text in the data availability section: "**6 Data availability**: The resulting European fuel map (circa 2019, 1 km spatial resolution) in one single-band categorical raster layer in GeoTIFF format is publicly available at https://doi.org/10.21950/YABYCN (Aragoneses et al., 2022a), as well as a Product User Manual (PUM) (Aragoneses et al., 2022b), at *e-cienciaDatos:* https://edatos.consorciomadrono.es/dataset.xhtml?persistentId=doi:10.21950/YABYCN."

References:
Aragoneses, E., Chuvieco, E., and García, M.: Product User Manual for the FirEurisk European fuel map, 1–9 pp., e-cienciaDatos, https://edatos.consorciomadrono.es/dataset.xhtml?persistentId=doi:10.21950/YABYCN, 2022b.

- Why the spatial resolution of the European fuel map classification is 1 km. Considering that most of the input layers used are between 100 and 300 m, I'm not sure why you reprojected the spatial resolution to 1 km. Maybe a reason could be the climate data that you were using from WorldClim2, or? In my opinion, if the main goal is to develop strategies on how handle forest fires at Pan European level the resolution of 1 km is quite wide for this goal.

Response: Although this 1 km resolution might be seen as coarse, it should be noted that the study area is the European continent. The extent of this study area and the available input data makes it difficult to map fuels at a finer scale, especially considering the input data (the CCI LC map at 300 m, the bioclimatic models (including the weather data), etcetera), which led to adopt 1 km as a compromise resolution. It should be considered that the intention to generate a European-scale fuel map is not to simulate fire propagation at regional or local scales, but to have an overview and a holistic perspective of the actual state of fuels in the European territory, that would help to identify areas with higher fire risk conditions and where specific prevention strategies might be adopted to reduce fire risk. In addition, the proposed fuel classification system is hierarchical and multipurpose, referring not only to propagation studies but also to emissions or post-fire recovery, for which 1 km resolution could be a first step to compare fuel and fire risk conditions at a continental scale. Anyway, the classification can be adapted to different spatial scales and in fact, the same legend is being used for Pilot Sites in Europe (regional scales at the FirEUrisk project where the target resolution is 1 Ha or higher), so it has a scalable structure. Nevertheless, similar methods could be developed to map fuel maps at higher resolution for the European scale in future works.

We have added the following to the manuscript in new line 349: "The input layers used for the generation of the European fuel map were previously resampled to 100 m to match the spatial resolution of the Copernicus GLC map, which was our main information source. However, the spatial characteristics of some of the input layers (such as the CCI LC map at 300 m, and the bioclimatic models based on 1 km resolution weather data), recommended to convert the final product to 1 km spatial resolution, which was also the project target resolution for the European scale. Therefore, after obtaining the first fuel type dataset at 100 m resolution, it was resampled to 1 km, carefully accounting for the heterogeneity of European fuel types."

- Regarding the validation, I'm not sure about the validation with another remote sensing input and not through ground truth data for the six main fuel types. However, if you are able to explain well the limitations of this work regarding this gap of information, it can be a valid approach for the two-validation process proposed.

Response: Although ideally validation should be based on field information, it was not feasible to perform an appropriate field validation campaign across Europe. Therefore, validation of the European fuel map was done using four information sources as reference data, from which one is the main source and the rest serve to complement when needed. The main information source, used for the two proposed validations, is LUCAS (Land Use and Coverage Area frame Survey) data from a similar period as the input data (year 2018). This data does not come from remote sensing, but instead they are georeferenced systematic field-surveyed points that allow to identify land cover and land use changes in the European Union including land use forms to fill on the field and photos of the surveyed points. Thus, this information source can be considered as ground truth data. Only when the LUCAS database was not enough to obtain all the information needed for the validation (and only for the second proposed validation), it was complemented with other information sources, namely Google Earth images, Google Street View images and the GlobLand30 map.

This is explained in the methods section in new line 376: "Considering the infeasibility of ground validation of the final product, we first validated the six main fuel types (forest, shrubland, grassland, cropland, wet and peat/semi-peat land, and urban) of our classification, plus the nonfuel category, using LUCAS (Land Use and Coverage Area frame Survey) as reference data. LUCAS points are derived from a field systematic survey, performed every three years by Eurostat to identify land cover and use changes (including photos) in the European Union (Eurostat, 2022a).", and for the second validation: "Since this required a visual interpretation, a 20 % subset of the 5,016 validation points was selected by stratified random sampling. Each point was assigned to a fuel category by visual interpretation of four information sources: 1) the 2018 LUCAS photos at a maximum distance of 200 m, 2) the latest Google Earth images to observe the 1 $km^2$ pixel, 3) Google Street View images, and 4) the 2020 global land cover GlobeLand30 map (30 m resolution) (Chen and Ban, 2014) with 85.72 % of overall accuracy, based on Landsat and Huanjing (HJ-1) images to help to validate forest and urban covers."

Results

The results are well addressed, but as I mentioned in the methodology section, it is difficult to validate a remote sensing product as a European fuel map classification with other remote sensing inputs. So, my main concern is the lack of field data in the validation. I'm aware of the limitations of the work at this scale. Therefore, these limitations should be properly addressed in the discussion.

Response: The reviewer is right in the fact that we did not visit the field for the validation. However, we did use field data to validate, as explained before. The main information source for validation were LUCAS points and photos, which are a widely used database for land cover mapping.

Discussion

The discussion needs to be addressed more realistic with the results obtained. Again, as I mentioned previously, I have my doubts if this fuel map describes also canopy fuels. Basically, there are two main concerns: i) fuelbed depth is just one variable (basically, the height of the vegetation), but other important attributes as the height at the crown interception and the crown bulk density are missing and ii) I have some doubts about the models that you were using in the fuelbed depth estimation. It is important to clarify this, if the main novelty in this work is the comparison from the previous one at global and European levels, is that these maps only dealt with surface fuels. In my opinion, the canopy fuels are not solved at all through the manuscript. Therefore, I would suggest addressing the discussion about the limitations of this work, that the canopy fuel still is a pending task, but the fuelbed depth is a contribution on it. Also, you can address the discussion on some recommendations on how canopy fuels can be produced in the future through remote sensing.

L110 Into the objectives, the first one is "generating a European advanced fuel classification system to facilitate the integration of continental wildfire risk assessment, including both surface and canopy fuels" I would like to highlight the last point. In my opinion, the canopy fuels are not really addressed within the methodology. It is true that you added the depth fuel (or height of the shrubland, grassland and so on). I have the feeling that canopy fuels is too ambitious according to the methodology and results. The depth fuel can partially be an input in the canopy fuels. However, other important metrics to understand better the canopy fuels are

missing, such as crown base height, crown bulk density, among others. Therefore, I would suggest being more focused on surface fuels, which is most of the work done through the manuscript.

Response: Regarding the comment on the results and the one for L110, we understand the reviewer's point, as there are some missing fuel parameters referring to canopy fuels, such as crown base height or crown bulk density. In this paper, we focus on the description of the legend of the proposed fuel classification system and mapping, and a first possibility to parameterize surface fuels using standard surface fuel models; and future works will focus on the parameterization of both canopy and surface fuels. Here, we mapped fuel types and fuelbed depth, which helps to identify fuel types that only differ in height. Some of the variables the reviewer proposes refer to fuel parameters, and this is subject of our on-going research Moreover, the proposed fuel typology includes different forest types because we understand that they present different canopy behaviour and, although their canopy parameters are not yet estimated, the fuel divisions are thought because they affect canopy fuels.

To clarify that there are canopy fuel parameters still missing, the objectives in new line 104 were modified as: "The first one was generating a fuel classification system to facilitate the integration of continental wildfire risk assessment, including both surface and canopy fuel types", and the following was added to the discussion in new line 716: "The full surface fuel set information needed to run fire propagation models can be extracted from the crosswalk to the FBFM, complemented with other canopy fuel parameters (such as crown base height or crown bulk density) and other necessary input data (e.g., weather conditions, topography, ignitions, etcetera) to run fire spread models (e.g., FlamMap (Finney, 2006) and FARSITE (Finney, 2004), as embedded in FlamMap 6.2 (https://www.firelab.org/project/flammap)."

Regarding the recommendations on how canopy fuels can be produced in the future through remote sensing, the discussion was completed with this in new line 720: "This should be subject of an extension of this paper and could be based on the calibration of models that estimate canopy fuel parameters using airborne and satellite LiDAR systems, for which regional airborne LiDAR would be key to consider the heterogeneity of European fuels before using the global satellite LiDAR data for the continental scale". Indeed, the authors are currently working on that, which is planned to be the next step in their research.

About the models used to derive the surface fuel heights, the discussion was modified to give a more realistic view of the results in new line 605: "Estimating shrubland and grassland fuelbed depth was challenging. To the best of our knowledge, there are no large-scale reliable datasets in Europe on these variables, which is limiting to our purposes. However, despite the models chosen to estimate surface fuelbed depth were not specifically developed for European areas, the biogeographical similarity of the regions for which they were developed to European conditions make them acceptable for our purposes."

Conclusions

The conclusions should be rewrite according to the previous suggestion and clearly define the novelty of the work compering to the previous one according to the limitations implied by a map at the European level. In short, the manuscript is a valuable contribution, but need to be improved before to be published according to the comments and suggestions provided.

Response: The conclusions were rewritten to:
1) Specify that the classification refers to both surface and canopy categories, but that the parameterization only referred to surface fuels.
2) Highlight the novelty aspects of the work.
3) Emphasize the limitations of the work, most of them related to the European level (size and complexity of the study area).

The conclusions are now as follows: "This paper, developed in the framework of the European FirEUrisk project, presents a new hierarchical fuel classification system for surface and canopy fuels adapted to the European conditions, as well as methods to map those categories and assign them fuel parameters. The final European fuel map contains 20 fuel types, including both surface and canopy fuel types. The estimated overall accuracy was 88 % for the main fuel types and 81 % for all mapped fuel types. Finally, the paper shows an example of a crosswalk between the proposed fuel types and standard fuel models, in this case the Fire Behaviour Fuel Models (FBFM) (Scott and Burgan, 2005), that provides a full set of surface fuel parameters useful for surface fire behaviour modelling. Our approach, based on expert knowledge, GIS, existing land cover datasets, biogeographic data, and bioclimatic modelling, could be readily applied to other regions.

The results of this study constitute the first step towards a risk-wise landscape and fuel mapping development across Europe, which will help integrated, strategic, coherent, and comprehensive decision making for fire risk prevention, assessment, and evaluation. The results have wide applicability because they meet the actual unfulfilled fuel mapping needs in Europe: 1) the development of a fuel classification system specifically designed for European conditions, which allows not to rely on external classifications that should be only applied to the regions for which they were developed, 2) enabling coordination, integrating fuel mapping at different spatial scales and across European regions through a common fuel legend with hierarchical levels, 3) multipurpose, including prevention, propagation, behaviour, emissions, and suppression, 4) mapping fuel types not previously considered at European scale that are key for protecting people and the environment from the devastating effects of fires: forest canopy fuels (key for crown and extreme fires), wet and peat/semi-peat land fuels (key for emissions) and urban fuels in the Wildland Urban Interface (key for people's and socio-economic safety), 5) the generation of an updated European-specific fuel map, compared to the EFFIS fuel map from year 2000 (European Forest Fire Information System (EFFIS), 2017), and 6) the preliminary surface fuel parameterization for Europe that can be used for estimating fuel parameters whenever there is no suitable input data available. Overall, the existence of updated land cover datasets and bioclimatic models for the European territory is limiting, and work is still needed to parameterize canopy fuels. The results of this work are part of the new FirEUrisk integrated three-part perspective of fire risk, whose strategy is meant to shift the thinking of wildfire management by looking simultaneously to fire assessment, reduction, and adaptation from a common scheme."

**Specific comments**

L20 In the abstract, you mentioned that the main fuel categories were grouped into six land uses, but I'm not sure why you included urban category, are you expecting forest fires into urban parks, or?

Response: Our proposed fuel classification system has the novelty of considering urban fuels, defined in this paper as those fuels in the intermix or interface of built-up areas, i.e. the Wildland Urban Interface (WUI). This constitutes an interesting improvement compared to pre-existing fuel classification systems where WUI areas are not considered fuels and therefore, are not parameterized (case of FBFM). Understanding urban fuels allows to assess residential and non-natural fuels, which can help to prevent fire risk to affecting human settlements and lives, and socio-economic losses (Bowman et al., 2017, 2020).

To clarify why urban fuels are included in the proposed classification, we have added the following to:

- New line 161: "Urban: areas with ≥ 15 % built-up structures and/or buildings. The standard CLC division between continuous and discontinuous fabric was followed, related to the amount of vegetation belonging to the intermix and interface of the Wildland-Urban Interface (WUI). This is part of the innovation of the proposed classification system, as it allows the assessment of residential and non-natural fuels, which can in turn help identifying anthropic areas where fires can affect human settlements and lives.",

- New line 568: "In addition, the FirEurisk fuel map includes new categories such as wet and peat/semi-peat land fuel types, which are key to understand fire emissions; and urban fuel types, crucial to prevent fire affecting humans, which were not considered in previous continental and global fuel maps.",

- New line 627: "Urban fuel types are the least represented in Europe, but they are the most dangerous from an economic, societal and human health point of view (Bowman et al., 2011). Mapping urban fuel types represents an advance of the proposed classification system, as it allows the assessment of residential and non-natural fuels, which can in turn help identifying anthropic areas where fires can affect human settlements and lives.",

- New line 770: "…4) mapping fuel types not previously considered at European scale that are key for protecting people and the environment from the devastating effects of fires: forest canopy fuels (key for crown and extreme fires), wet and peat/semi-peat land fuels (key for emissions) and urban fuels in the Wildland Urban Interface (key for people's and socio-economic safety)…".

References:

Bowman, D., Williamson, G., Yebra, M., Lizundia-Loiola, J., Pettinari, M. L., Shah, S., Bradstock, R., and Chuvieco, E.: Wildfires: Australia needs national monitoring agency, Nature, 584 (7820), 188–191, https://doi.org/10.1038/d41586-020-02306-4, 2020.

Bowman, D. M. J. S., Williamson, G. J., Abatzoglou, J. T., Kolden, C. A., Cochrane, M. A., and Smith, A. M. S.: Human exposure and sensitivity to globally extreme wildfire events, Nat. Ecol. Evol., 1, https://doi.org/10.1038/s41559-016-0058, 2017.

L145 You defined the vegetation height as fuel depth, but I'm not sure if this term is correct, maybe "fuel height"?

Response: Here we are referring to the fuelbed depth, which is associated to surface fuels. We have changed "fuel depth" by "fuelbed depth" in the text to indicate that we are referring to surface fuels, including shrubland and grassland. This terminology is largely accepted (see https://www.nwcg.gov/term/glossary/fuel-bed-depth). Also, this was clarified the first time the term appears in the text in new line 175: "For shrubland and grassland fuel types, subcategories were created based on fuelbed depth (height of the surface fuel layer)."

L160 Garrigue and maquis are just terms used in different countries, but with the same meaning "shrubland". I would suggest deleting these terms because is repetitive or explain that is a specific term according to a specific country.

Response: According to the reviewer's suggestion, the two terms were deleted from the definition of shrubland.

L195 Since the article probably will be published in 2023, it could be important also highlight the recent forest fire between the Czech Republic and Germany in 2022 and how the forest fires are going more up in latitude.

Response: As suggested by the reviewer, this information was added in new line 194: "However, a recent increase in fire activity in higher latitudes has been observed: e.g., fires in Sweden in 2018 (San-Miguel-Ayanz et al., 2021), and the fire between the Czech Republic and Germany in 2022 (Global Disaster Alert and Coordination system, 2022)."

References:

Global Disaster Alert and Coordination system: Forest Fire (2287 ha) in Germany, Czech Republic 24 Jul 2022, 2022, https://www.gdacs.org/report.aspx?eventtype=WF&eventid=1007792, last access 18 November 2022.

L230 It is not clear at all to me how the fuel height was determined and how accurate it was considering that the units are in centimetres, which is hard to handle from earth observation data (and models). Are some ground truth data available to validate the fuelbed depth?

Response: The fuel height was estimated using the bioclimatic models.
Regarding the first part of the comment, the authors are aware that the estimation in centimetres can be tricky, and its accuracy is limited. That is why the proposed classification and the final fuel map aggregate the fuel types in three groups: high, medium, and low fuelbed depth, not giving specific heights in cm.
Regarding the second part of the comment, these models are already calibrated and validated, and their details can be read in the papers where they were originally presented. However, the authors also validated the results of their application in the validation section of the fuel map (by using the LUCAS photos). LUCAS photos do not come from remote sensing but are a field survey, as explained before, which is believed to serve for our validation purposes. However, the authors are aware about the limitations of estimating and validating fuelbed depth because of its high temporal variability. This is the reason why the bioclimatic models refer to the potential fuelbed depth. Also, this has been highlighted in the discussion in new line 678: "Shrubland and grassland fuel types' errors are significant, mostly between fuelbed depth categories. Therefore, care must be taken for these results, as estimating fuelbed depth from photos is challenging, and fuelbed depth varies with

time. These limitations specially affect grassland due to its low depth, rapid growth, and that high grassland is frequently cut. Thus, grassland fuelbed depth is very changeable so we assume the European fuel map may only be accurate for some periods of the year. We validated the proposed fuel map considering the mean potential fuelbed depth."

L245 Considering that the map layers are from different years (e.g., Copernicus GLC map 2019, CCI LC map 2020, etc.) I would suggest adding the years when the layers were created in the workflow of the figure 3.

Response: Done. The new Figure is as follows:

[Figure]

**Figure 3.** Methodology used to generate the European fuel map. The input sources are in the text.

L290 Where the biomass data is coming from to feed the depth (m) response variable? I have my doubts about the models, especially in its validation (see comment in L230). The same concern is in equation 3.

Response: The biomass data used in equation 2 is calculated in equation 1. For shrubland, this is a two-steps model: first, Mean Annual Precipitation (MAP) data is used to estimate biomass; second, the estimated biomass is used to estimate the potential fuelbed height. The reasoning behind this model is that shrubland depth is directly related to shrubland productivity (Radloff and Mucina, 2007; Saglam et al., 2008; Ali et al., 2015), which is mainly determined by the Mean Annual Precipitation (MAP) (Shoshany and Karnibad, 2015; Paradis et al., 2016; Bohlman et al., 2018; Zhang et al., 2018b) through biomass accumulation (Keeley and Keeley, 1977; Schlesinger and Gill, 1980; Gray and Schlesinger, 1981; Bohlman et al., 2018).
For grassland, the model has only one step: grassland fuelbed depth was directly estimated from grassland biomass data. Again, the reasoning behind this is that grassland depth is directly related to grassland productivity or biomass (Zhang et al., 2018a; Crabbe et al., 2019; Michez et al., 2019; Batistoti et al., 2019). For this case, we directly used already-existing information on mean grassland biomass for European zones based on consistent inventory of regional statistics on this variable (Smit et al., 2008).

References:
Ali, A., Xu, M.-S., Zhao, Y.-T., Zhang, Q.-Q., Zhou, L.-L., Yang, X.-D., and Yan, E.-R.: Allometric biomass equations for shrub and small tree species in subtropical China, Silva Fenn., 49 (4), https://doi.org/10.14214/sf.1275, 2015.
Batistoti, J., Marcato, J., Ítavo, L., Matsubara, E., Gomes, E., Oliveira, B., Souza, M., Siqueira, H., Filho, G. S., Akiyama, T., Gonçalves, W., Liesenberg, V., Li, J., and Dias, A.: Estimating Pasture Biomass and Canopy Height in Brazilian Savanna Using UAV Photogrammetry, Remote Sens., 11 (20), 2447, https://doi.org/10.3390/RS11202447, 2019.
Bohlman, G. N., Underwood, E. C., and Safford, H. D.: Estimating Biomass in California's Chaparral and

Coastal Sage Scrub Shrublands, Madroño, 65 (1), 28–46, https://doi.org/10.3120/0024-9637-65.1.28, 2018.

Crabbe, R. A., Lamb, D. W., Edwards, C., Andersson, K., and Schneider, D.: A preliminary investigation of the potential of Sentinel-1 radar to estimate pasture biomass in a grazed pasture landscape, Remote Sens., 11 (7), 872, https://doi.org/10.3390/RS11070872, 2019.

Gray, J. T. and Schlesinger, W. H.: Biomass, production, and litterfall in the coastal sage scrub of Southern California, Am. J. Bot., 68 (1), 24–33, https://doi.org/10.1002/J.1537-2197.1981.TB06352.X, 1981.

Keeley, J. E. and Keeley, S. C.: Energy Allocation Patterns of a Sprouting and a Nonsprouting Species of Arctostaphylos in the California Chaparral, Am. Midl. Nat., 98 (1), 1–10, https://doi.org/10.2307/2424710, 1977.

Michez, A., Lejeune, P., Bauwens, S., Lalaina Herinaina, A. A., Blaise, Y., Muñoz, E. C., Lebeau, F., and Bindelle, J.: Mapping and monitoring of biomass and grazing in pasture with an unmanned aerial system, Remote Sens., 11 (5), 476, https://doi.org/10.3390/RS11050473, 2019.

Paradis, M., Lévesque, E., and Boudreau, S.: Greater effect of increasing shrub height on winter versus summer soil temperature, Environ. Res. Lett., 11 (8), 085005, https://doi.org/10.1088/1748-9326/11/8/085005, 2016.

Radloff, F. G. T. and Mucina, L.: A quick and robust method for biomass estimation in structurally diverse vegetation, J. Veg. Sci., 18 (5), 719–724, https://doi.org/10.1111/J.1654-1103.2007.TB02586.X, 2007.

Saglam, B., Küçük, Ö., Bilgili, E., and Durmaz, B. D.: Estimating Fuel Biomass of Some Shrub Species (Maquis) in Turkey, Turkish J. Agric. For., 32, 349–356, 2008.

Schlesinger, W. H. and Gill, D. S.: Biomass, Production, and Changes in the Availability of Light, Water, and Nutrients During the Development of Pure Stands of the Chaparral Shrub, Ceanothus Megacarpus, After Fire, Ecology, 61 (4), 781–789, https://doi.org/10.2307/1936748, 1980.

Shoshany, M. and Karnibad, L.: Remote sensing of shrubland drying in the South-East Mediterranean, 1995-2010: Water-use-efficiency-based mapping of biomass change, Remote Sens., 7 (3), 2283–2301, https://doi.org/10.3390/RS70302283, 2015.

Smit, H. J., Metzger, M. J., and Ewert, F.: Spatial distribution of grassland productivity and land use in Europe, Agric. Syst., 98 (3), 208–219, https://doi.org/10.1016/j.agsy.2008.07.004, 2008.

Zhang, H., Sun, Y., Chang, L., Qin, Y., Chen, J., Qin, Y., Du, J., Yi, S., and Wang, Y.: Estimation of grassland canopy height and aboveground biomass at the quadrat scale using unmanned aerial vehicle, Remote Sens., 10 (6), 1–19, https://doi.org/10.3390/rs10060851, 2018a.

Zhang, X., Guan, T., Zhou, J., Cai, W., Gao, N., Du, H., Jiang, L., Lai, L., and Zheng, Y.: Community Characteristics and Leaf Stoichiometric Traits of Desert Ecosystems Regulated by Precipitation and Soil in an Arid Area of China, Int. J. Environ. Res. Public Health, 15 (1), 109, https://doi.org/10.3390/IJERPH15010109, 2018b.

L420 The histogram showed shrubs up to 6 m. I believe that there is a misunderstanding in the definition between what is a tree and what is a shrub. Probably, in the distribution that the histogram is showing, there are some trees included.

Response: The following in new line 430 was added and is self-explicative: "Although shrubland are generally considered up to 5 m, exceptions are allowed subject to the plant's physiognomic aspect (Food and Agriculture Organization, 2000). Therefore, here we allowed for plants higher to 5 m being classified as shrubland if they have a clear physiognomic aspect of shrub."

L455 The table 4 is quite large and the same information is repeated in the text. Maybe, it would be better to replace the table by pie graphs or send as supplementary material.

L525 I would suggest the same that in L455, the table 8 is quite large and could be more suitable transform in a pie graph only the % values.

Response: Following the reviewer's suggestion, Tables 4 and 8 were sent to Appendices. Pie graphs were not included as they were considered to be repetitive with the information already provided in the paper: Tables in the Appendix, fuel maps and description in the text.

L545 Figure 7 is connected to figure 4? Because the results are completely different but is the same attributed measured. Or?

Response: These two figures show different things. On one hand, Figure 4 represents the fuelbed dept for every shrubland and grassland pixel after applying the bioclimatic models. These pixels where then aggregated into the three height groups (high, medium, and low) according to the proposed thresholds, resulting in part of Figure 5. On the other hand, Figure 7 is the result of applying the crosswalk from the proposed fuel types to the FBFM models. Each of the FBFM models has already-defined fuel parameters, which can be mapped after applying the crosswalk to the European territory. In this sense, Figure 7 shows an example of mapped parameters from the application of the crosswalk. The fuelbed depth in this last case refers only to surface fuels and refers to it in shrubland, grassland and forest environments, whereas Figure 4 only refers to the pixels identified as shrubland and grassland in the European fuel map for the FirEUrisk fuel classification system.

To clarify this in the text, some modifications have been done:

"**Figure 4.** Histograms for shrubland and grassland fuelbed depth (m) in Europe obtained from the application of the bioclimatic models. The blue lines represent the fuelbed depth threshold used to subdivide shrubland and grassland fuel types."

"**Figure 7.** Surface dead 1h fuel load and fuelbed depth over Europe obtained from the crosswalk from the FirEUrisk fuel types to the FBFM models. Note that surface fuelbed depth for the forest fuels refers to the understory, not the crowns."

Please, find also attached the new version of the manuscript with the changes done.